# InfiFPO: Implicit Model Fusion via Preference Optimization in Large Language Models

**Yanggan Gu**[1,2]     **Yuanyi Wang**[1]     **Zhaoyi Yan**[2]     **Yiming Zhang**[1]
**Qi Zhou**[1]     **Fei Wu**[3]     **Hongxia Yang**[1,2*]
[1]The Hong Kong Polytechnic University     [2]InfiX.ai     [3]Zhejiang University
yanggangu@outlook.com   hongxia.yang@polyu.edu.hk

Project Page: https://github.com/InfiXAI/InfiFPO

## Abstract

Model fusion combines multiple Large Language Models (LLMs) with different strengths into a more powerful, integrated model through lightweight training methods. Existing works on model fusion focus primarily on supervised fine-tuning (SFT), leaving preference alignment (PA) —a critical phase for enhancing LLM performance—largely unexplored. The current few fusion methods on PA phase, like WRPO, simplify the process by utilizing only response outputs from source models while discarding their probability information. To address this limitation, we propose **InfiFPO**, a preference optimization method for implicit model fusion. InfiFPO replaces the reference model in Direct Preference Optimization (DPO) with a fused source model that synthesizes multi-source probabilities at the sequence level, circumventing complex vocabulary alignment challenges in previous works and meanwhile maintaining the probability information. By introducing probability clipping and max-margin fusion strategies, InfiFPO enables the pivot model to align with human preferences while effectively distilling knowledge from source models. Comprehensive experiments on 11 widely-used benchmarks demonstrate that InfiFPO consistently outperforms existing model fusion and preference optimization methods. When using Phi-4 as the pivot model, InfiFPO improve its average performance from 79.95 to 83.33 on 11 benchmarks, significantly improving its capabilities in mathematics, coding, and reasoning tasks.

## 1   Introduction

Large Language Models (LLMs) have demonstrated impressive capabilities across a wide range of natural language tasks. Yet, no single model is universally optimal—different LLMs often possess distinct strengths due to variations in architecture, pretraining data, and objectives. This has motivated a surge of interest in model fusion, which aims to integrate knowledge from multiple source models into a single pivot model to enhance its overall performance [1–5].

While existing work on model fusion has primarily focused on the supervised fine-tuning (SFT) phase, little attention has been paid to integrating fusion techniques into the preference alignment phase, a critical step in reinforcement learning from human feedback (RLHF) pipelines that substantially boosts performance and usability. Applying model fusion to this phase is non-trivial due to the discrete nature of preference data and the challenge of aligning both preferred and dispreferred outputs across heterogeneous models.

The closest prior work, WRPO [6], attempts to bridge this gap by incorporating high-quality source model responses as additional preference signals. However, WRPO suffers from two major limitations:

---

*Corresponding author.

39th Conference on Neural Information Processing Systems (NeurIPS 2025).

1) it discards the probabilistic outputs of source models and only uses response-level supervision; and 2) it focuses solely on preferred responses, missing valuable contrastive signals for dispreferred ones. As a result, WRPO only partially leverages the capabilities of the source models. The broader question—*how to systematically fuse model knowledge during preference alignment*—remains largely unanswered.

To overcome these limitations, we propose **InfiFPO**, a principled and efficient framework for performing model fusion during the preference alignment phase. Our key insight is that the reference model in preference optimization (e.g., in DPO) can be replaced with a fused source model, thereby enabling the pivot model to learn not only from preference data but also from the probabilistic behaviors of multiple source models. Unlike WRPO, InfiFPO utilizes sequence probability from all source models—including for both preferred and dispreferred responses—making the fusion process more principled and information-rich. We call this fusion that utilizes sequence-level probabilities as *implicit model fusion*, distinguishing it from previous work on token-level model fusion.

To instantiate InfiFPO efficiently, we derive it from an RLHF-style constrained optimization framework called FuseRLHF, which encourages the pivot model to maximize preference rewards while remaining close—in sequence-level KL divergence—to each source model. We further transform this constrained RL problem into a fully offline optimization objective that avoids expensive online sampling and reward model training.

To improve the robustness and effectiveness of InfiFPO, we introduce three key enhancements: (1) **Length Normalization**, which reduces bias arising from varying token sequence lengths across models; (2) **Probability Clipping**, which limits the influence of underperforming source models by suppressing noisy gradients; and (3) **Max-Margin Fusion**, which adaptively prioritizes source models that offer the most distinctive and informative deviations from the pivot model.

We conducted a comprehensive evaluation of InfiFPO using Phi-4 [7] as the pivot model and selecting five mainstream open-source LLMs with parameters ranging from 9B to 24B as source models. Our experiments spanned 11 widely-used benchmarks covering diverse tasks, including mathematics, coding, instruction following, and so on. Results demonstrate that InfiFPO consistently outperforms existing model fusion and preference optimization methods, improving Phi-4's average performance from 79.95 to 83.33 across 11 benchmarks. Furthermore, InfiFPO exhibits great versatility, effectively combining with various preference optimization objectives to further enhance performance.

In summary, our contributions are threefold:
❶ We propose **InfiFPO**, a novel preference optimization framework that performs implicit model fusion by replacing the reference model in DPO with a fused source model, derived via an offline relaxation of a sequence-KL constrained RLHF objective.
❷ We introduce three stability-enhancing strategies—**length normalization**, **probability clipping**, and **max-margin fusion**—to avoid degradation from tokenization mismatch and probability inconsistencies in source models.
❸ We conduct extensive experiments on 11 preference benchmarks, demonstrating that InfiFPO consistently and significantly outperforms existing model fusion and preference optimization baselines, verifying its effectiveness for implict model fusion[2].

## 2   Preliminaries

In this section we briefly revisit the two foundations of our work: *Model Fusion* [1, 2, 4] and *Direct Preference Optimization* (DPO)[8].

**Model Fusion**: The goal of model fusion is to integrate knowledge from source models with different architectures, parameter sizes, and training data into one pivot model to enhance the pivot model's capabilities. Given $N$ source models $\{\mathcal{M}_i^s\}_{i=1}^N$ and a pivot model (also called target model) $\mathcal{M}^p$, the model fusion can be cast as a KL-constrained optimization problem:

$$\arg\max_{\mathcal{M}^p} \mathbb{E}_{\boldsymbol{x},\boldsymbol{y}\sim\mathcal{D}}\left[\log\mathcal{M}^p(\boldsymbol{y}|\boldsymbol{x})\right]$$
$$\text{s.t. } \mathbb{E}_{\boldsymbol{x},\boldsymbol{y}\sim\mathcal{D}}\left[\mathbb{D}_{\text{TKL}}\left[\mathcal{M}_i^s(\boldsymbol{y}|\boldsymbol{x})||\mathcal{M}^p(\boldsymbol{y}|\boldsymbol{x})\right]\right] \leq \varepsilon, \quad \forall i \in \{1,...,N\}, \tag{1}$$

---

[2]Our project is available at https://github.com/Reallm-Labs/InfiFPO.

where each sample in the dataset $\mathcal{D}$ contains a prompt sequence $\boldsymbol{x}$ and its corresponding response sequence $\boldsymbol{y}$. $\mathbb{D}_{\mathrm{TKL}}$ indicates token-level KL divergence. Each constraint here keeps the pivot model within an $\varepsilon$-ball of every source model, thereby fusing their behaviors. While effective, Eq. (1) suffers from **vocabulary conflict**: the source models often employ incompatible tokenisers, forcing previous work to rely on heuristic vocabulary matching.

**Direct Preference Optimization**: DPO [8] is an offline replacement for Reinforcement Learning from Human Feedback (RLHF). The objective of RLHF can be formalized as

$$\arg\max_{\mathcal{M}^\theta} \mathbb{E}_{\boldsymbol{x}\sim\mathcal{D},\boldsymbol{y}\sim\mathcal{M}^\theta(\boldsymbol{y}|\boldsymbol{x})}[\boldsymbol{r}(\boldsymbol{x},\boldsymbol{y})] - \beta\mathbb{D}_{\mathrm{SKL}}\left[\mathcal{M}^\theta(\boldsymbol{y}|\boldsymbol{x})\|\mathcal{M}^{\mathrm{ref}}(\boldsymbol{y}|\boldsymbol{x})\right]. \tag{2}$$

where $\boldsymbol{r}$ is a reward model that evaluates how good the response $\boldsymbol{y}$ generated by policy model $\mathcal{M}^\theta$ is. $\mathbb{D}_{\mathrm{SKL}}$ indicates sequence-level KL divergence[3]. Usually, the base reference model $\mathcal{M}^{\mathrm{ref}}$ is the initial $\mathcal{M}^\theta$, and $\beta$ is a parameter controlling the deviation from $\mathcal{M}^{\mathrm{ref}}$.

After deriving Eq. (2), the relationship between $\boldsymbol{r}$ and the optimal policy $\mathcal{M}^{\theta*}$ is as follows

$$\boldsymbol{r}(\boldsymbol{x},\boldsymbol{y}) = \beta\log\frac{\mathcal{M}^{\theta*}(\boldsymbol{y}|\boldsymbol{x})}{\mathcal{M}^{\mathrm{ref}}(\boldsymbol{y}|\boldsymbol{x})} + \beta\log Z(\boldsymbol{x}). \tag{3}$$

where $Z(\boldsymbol{x}) = \sum_{\boldsymbol{y}} \mathcal{M}^{\mathrm{ref}}(\boldsymbol{y}|\boldsymbol{x})) \exp\left(\frac{1}{\beta}\boldsymbol{r}(\boldsymbol{x},\boldsymbol{y}))\right)$.

Since the reference model $\mathcal{M}^{\mathrm{ref}}$ remains unchanged during training, the optimal policy model is theoretically determined only by the reward model $\boldsymbol{r}$. A natural idea is to indirectly train the optimal policy model through training the reward model, which is the optimization objective of DPO:

$$\mathcal{L}_{\mathrm{DPO}}(\mathcal{M}^\theta;\mathcal{M}^{\mathrm{ref}}) = -\mathbb{E}_{(\boldsymbol{x},\boldsymbol{y}_w,\boldsymbol{y}_l)\sim\mathcal{D}^p}\left[\log\sigma\left(\beta\log\frac{\mathcal{M}^\theta(\boldsymbol{y}_w|\boldsymbol{x})}{\mathcal{M}^{\mathrm{ref}}(\boldsymbol{y}_w|\boldsymbol{x})} - \beta\log\frac{\mathcal{M}^\theta(\boldsymbol{y}_l|\boldsymbol{x})}{\mathcal{M}^{\mathrm{ref}}(\boldsymbol{y}_l|\boldsymbol{x})}\right)\right]. \tag{4}$$

where each sample in the preference dataset $\mathcal{D}^p$ contains a prompt $\boldsymbol{x}$, a preferred response $\boldsymbol{y}_w$, and a dispreferred response $\boldsymbol{y}_l$.

# 3 InfiFPO: Preference Optimization for Model Fusion

This section presents InfiFPO, our novel approach to preference optimization for Model Fusion. We begin by introducing the FuseRLHF objective (§ 3.1), which integrates model fusion with RLHF. Given the inherent complexity of reinforcement learning frameworks, direct implementation proves challenging. Consequently, we propose an efficient offline InfiFPO methodology and develop three performance-enhancing strategies (§ 3.2). Finally, we provide gradient analysis to further understand the optimization mechanism of InfiFPO (§ 3.3)

## 3.1 FuseRLHF: RLHF for Implicit Model Fusion

**Constrained objective.** We consider model fusion during the preference alignment phase to be a constrained optimization problem. Based on Eq. (1) and (2), we can obtain the objective of FuseRLHF:

$$\arg\max_{\mathcal{M}^p} \mathbb{E}_{\boldsymbol{x},\boldsymbol{y}\sim\mathcal{D}}\left[\boldsymbol{r}(\boldsymbol{x},\boldsymbol{y})\right]$$
$$\text{s.t. } \mathbb{E}_{\boldsymbol{x},\boldsymbol{y}\sim\mathcal{D}}\left[\mathbb{D}_{\mathrm{SKL}}\left[\mathcal{M}_i^s(\boldsymbol{y}|\boldsymbol{x})\|\mathcal{M}^p(\boldsymbol{y}|\boldsymbol{x})\right]\right] \leq \varepsilon, \quad \forall i \in \{1,...,N\}, \tag{5}$$

where the initial pivot model is included in the source models. Each constraint keeps the pivot policy within an $\varepsilon$-ball (in KL divergence) of every teacher, thereby fusing their behaviour while still allowing preference alignment through the reward $\boldsymbol{r}$.

Please note that we use KL constraints at the sequence-level instead of the token-level. As shown in Eq. (1), previous works on model fusion used token-level KL and faced the vocabulary conflict

---

[3]Due to the discrete nature of language generation, this function is not differentiable and usually be simplified as $\mathbb{D}_{\mathrm{SKL}}\left[\mathcal{M}^\theta(\boldsymbol{y}|\boldsymbol{x})\|\mathcal{M}^{\mathrm{ref}}(\boldsymbol{y}|\boldsymbol{x})\right] = \log(\mathcal{M}^\theta(\boldsymbol{y}|\boldsymbol{x})) - \log(\mathcal{M}^{\mathrm{ref}}(\boldsymbol{y}|\boldsymbol{x}))$.

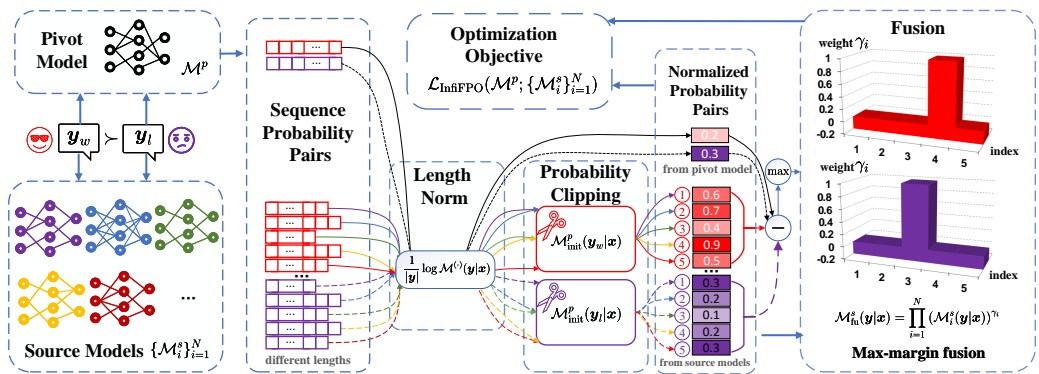

Figure 1: Overview of InfiFPO for implicit model fusion. We compute probabilities for preferred ($\boldsymbol{y}_w$) and dispreferred ($\boldsymbol{y}_l$) responses using both pivot and source models. Following length normalization and probability clipping, we identify the source model with the maximum normalized probability difference from the pivot model for fusion and preference alignment.

issue, where heterogeneous models have different vocabularies, making their token-level output distributions incompatible for divergence calculation. To solve this issue, those works introduce complex vocabulary matching processes and only calculate KL divergence on the matching portions. In contrast, our sequence-level KL constraints avoid this issue while preserving the source models' probability information, learning source models' preferences for the whole sentence. We call this fusion that utilizes sequence-level probabilities as **implicit model fusion** (IMF), distinguishing it from previous work on token-level model fusion.

**Unconstrained objective.** Introducing non-negative multipliers $\{\gamma_i\}_{i=1}^N$ and relaxing the constraints in Eq. (5) yields the following unconstrained objective:

$$\underset{\mathcal{M}^p}{\arg\max} \ \mathbb{E}_{\boldsymbol{x}\sim\mathcal{D},\boldsymbol{y}\sim\mathcal{M}^p(\boldsymbol{y}|\boldsymbol{x})}[\boldsymbol{r}(\boldsymbol{x},\boldsymbol{y})] - \beta \sum_{i}^{N} \gamma_i \mathbb{D}_{\mathrm{SKL}}\left[\mathcal{M}^p(\boldsymbol{y}|\boldsymbol{x})\|\mathcal{M}_i^s(\boldsymbol{y}|\boldsymbol{x})\right], \quad (6)$$

where $\gamma_i \geq 0$ and $\sum_{i=1}^N \gamma_i = 1$. By setting $\gamma_i$ to different values, various fusion strategies can be adopted. Specifically, when $N = 1$ and taking the source model as the initial pivot model, Eq. (14) reduces to classical RLHF.

## 3.2 InfiFPO: Efficient Preference Optimization for IMF

**Deriving the FPO objective.** Directly training the pivot model with FuseRLHF requires significant time and resources. On one hand, it requires training an additional reward model; on the other hand, during RL, the pivot model needs to sample responses online, while all source models also need to calculate sequence probabilities simultaneously. To reduce these costs, we follow Rafailov et al. [8] by converting online FuseRLHF to offline infiFPO, improving training efficiency. Following prior works, it is clear to show that the optimal pivot model $\mathcal{M}^{p*}$ to the FuseRLHF objective in Eq. (14) takes the form:

$$\mathcal{M}^{p*}(\boldsymbol{y}|\boldsymbol{x}) = \frac{1}{Z(x)}\mathcal{M}_{\mathrm{fu}}^s(\boldsymbol{y}|\boldsymbol{x})\exp(\frac{1}{\beta}\boldsymbol{r}(\boldsymbol{x},\boldsymbol{y})). \quad (7)$$

where a fused source model $\mathcal{M}_{\mathrm{fu}}^s(\boldsymbol{y}|\boldsymbol{x}) = \prod_{i=1}^N (\mathcal{M}_i^s(\boldsymbol{y}|\boldsymbol{x}))^{\gamma_i}$ and a partition function $Z(\boldsymbol{x}) = \sum_{\boldsymbol{y}} \mathcal{M}_{\mathrm{fu}}^s(\boldsymbol{y}|\boldsymbol{x})\exp\left(\frac{1}{\beta}\boldsymbol{r}(\boldsymbol{x},\boldsymbol{y})\right)$. See Appendix A.1 for a complete derivation. Then we can rearrange Eq. (7) to show the relationship between the reward model and the optimal pivot model as:

$$\boldsymbol{r}(\boldsymbol{x},\boldsymbol{y}) = \beta \log \frac{\mathcal{M}^{p*}(\boldsymbol{y}|\boldsymbol{x})}{\mathcal{M}_{\mathrm{fu}}^s(\boldsymbol{y}|\boldsymbol{x})} + \beta \log Z(\boldsymbol{x}). \quad (8)$$

We can see that theoretically the optimal pivot model $\mathcal{M}^{p*}$ only depends on the reward model $\boldsymbol{r}$, since the source models remain unchanged during training. Therefore, we can derive the optimization

objective of FPO as follows:

$$\mathcal{L}_{\text{FPO}}(\mathcal{M}^p; \{\mathcal{M}_i^s\}_{i=1}^N) = -\mathbb{E}_{(\boldsymbol{x},\boldsymbol{y}_w,\boldsymbol{y}_l)\sim\mathcal{D}^p}\left[\log\sigma\left(\beta\log\frac{\mathcal{M}^p(\boldsymbol{y}_w|\boldsymbol{x})}{\mathcal{M}_{\text{fu}}^s(\boldsymbol{y}_w|\boldsymbol{x})} - \beta\log\frac{\mathcal{M}^p(\boldsymbol{y}_l|\boldsymbol{x})}{\mathcal{M}_{\text{fu}}^s(\boldsymbol{y}_l|\boldsymbol{x})}\right)\right].\tag{9}$$

Especially, when $N = 1$ and using the initial pivot model as the source model, this loss function reduces to the original DPO, i.e., Eq. (4). While the FPO objective is simple and theoretically supported, it faces *length normalization* and *source model degradation* issues in model fusion. Besides, the fusion multipliers $\{\gamma_i\}_{i=1}^N$ for source models requires a strategy for determination. To address these issues, we propose the following three strategies.

**Length Normalization.** Models with larger vocabularies tend to produce longer segmentations whose log-likelihood sums are lower, even when the semantic adequacy of the response is unchanged. Similar effects have been presented by Wu et al. [9]. To address the length bias issue, we introduce the length normalization, which divides the sequence log probability from a model by the length of the sequence after tokenization with that model.

$$\overline{\log}\,\mathcal{M}^{(\cdot)}(\boldsymbol{y}|\boldsymbol{x}) = \frac{1}{|\boldsymbol{y}|}\log\mathcal{M}^{(\cdot)}(\boldsymbol{y}|\boldsymbol{x})\tag{10}$$

where $\mathcal{M}^{(\cdot)}$ can be either a source model or a pivot model.

**Probability Clipping.** Source models may occasionally assign lower probability to the preferred response $\boldsymbol{y}_w$ (or higher probability to the dispreferred response $\boldsymbol{y}_l$) than the pivot model, injecting misleading gradients and causing instability. To avoid the source model degradation issue, we clip the sequence probabilities of the source model. Specifically, for $\boldsymbol{y}_w$, we define the minimum probability of the source model as the sequence probability of the initial pivot model $\mathcal{M}_{\text{init}}^p$; for $\boldsymbol{y}_l$, we define the maximum sequence probability of the source model as the sequence probability of the initial pivot model.

$$\text{Clip}(\mathcal{M}_i^s(\boldsymbol{y}|\boldsymbol{x})) = \begin{cases}\max(\mathcal{M}_i^s(\boldsymbol{y}|\boldsymbol{x}), \mathcal{M}_{\text{init}}^p(\boldsymbol{y}|\boldsymbol{x})), & \text{if } \boldsymbol{y} \text{ is } \boldsymbol{y}_w,\\ \min(\mathcal{M}_i^s(\boldsymbol{y}|\boldsymbol{x}), \mathcal{M}_{\text{init}}^p(\boldsymbol{y}|\boldsymbol{x})), & \text{else.}\end{cases}\tag{11}$$

This piecewise definition is weakly monotone: for all $t_1 \leq t_2$ we have $\text{Clip}(t_1) \leq \text{Clip}(t_2)$, so the logarithm that follows preserves (or ties) the order of probabilities and cannot invert preferences.

**Max-margin Fusion.** To determine the multipliers $\{\gamma_i\}_{i=1}^N$ for source models, we propose a max-margin fusion strategy that maximizes the diversity of information incorporated into the pivot model. Specifically, we select the source model that differs most from the current pivot model, as this model likely contains the most unique and complementary information.

$$\gamma_i = \begin{cases}1, & \text{if } i = \arg\max_j\ \text{D}(\mathcal{M}_j^s, \mathcal{M}^p),\\ 0, & \text{otherwise.}\end{cases}\tag{12}$$

where $\text{D}(\mathcal{M}_j^s, \mathcal{M}^p) = \text{abs}(\mathcal{M}_j^s(\boldsymbol{y}|\boldsymbol{x}) - \mathcal{M}^p(\boldsymbol{y}|\boldsymbol{x}))$ measures the probability difference between a source model and the pivot model. This winner-takes-all approach simplifies training while effectively capturing diverse capabilities across source models. In §4.3, we also evaluate alternative fusion strategies, including averaging and dynamic weighting methods.

**In Summary.** Combining the strategies listed above, we have the final InfiFPO objective:

$$\mathcal{L}_{\text{InfiFPO}}(\mathcal{M}^p; \{\mathcal{M}_i^s\}_{i=1}^N) = -\mathbb{E}_{(\boldsymbol{x},\boldsymbol{y}_w,\boldsymbol{y}_l)\sim\mathcal{D}^p}\left[\log\sigma\left(\beta\overline{\log}\frac{\mathcal{M}^p(\boldsymbol{y}_w|\boldsymbol{x})}{\mathcal{M}_{\text{fu}}^{sclip}(\boldsymbol{y}_w|\boldsymbol{x})} - \beta\overline{\log}\frac{\mathcal{M}^p(\boldsymbol{y}_l|\boldsymbol{x})}{\mathcal{M}_{\text{fu}}^{sclip}(\boldsymbol{y}_l|\boldsymbol{x})}\right)\right],\tag{13}$$

where $\mathcal{M}_{\text{fu}}^{sclip}(\boldsymbol{y}|\boldsymbol{x}) = \prod_{i=1}^N\left[\text{Clip}(\mathcal{M}_i^s(\boldsymbol{y}|\boldsymbol{x}))\right]^{\gamma_i}$.

### 3.3 Gradient Analysis.

To gain deeper insights into InfiFPO's optimization dynamics, we analyze the gradient of the loss function with respect to the pivot model parameters $\theta$. This analysis elucidates how information from source models influences the pivot model's learning trajectory. The gradient can be expressed as:

$$\nabla\theta\mathcal{L}_{\text{InfiFPO}} =$$

$$-\beta\mathbb{E}_{(\boldsymbol{x},\boldsymbol{y}_w,\boldsymbol{y}_l)\sim\mathcal{D}^p}\left[\underbrace{\sigma\left(\beta\overline{\log}R_s - \beta\overline{\log}R_p\right)}_{\text{preference disparity coefficient}}\left[\underbrace{\nabla\theta\overline{\log}\mathcal{M}^p(\boldsymbol{y}_w|\boldsymbol{x})}_{\text{increase likelihood of } \boldsymbol{y}_w} - \underbrace{\nabla\theta\overline{\log}\mathcal{M}^p(\boldsymbol{y}_l|\boldsymbol{x})}_{\text{decrease likelihood of } \boldsymbol{y}_l}\right]\right].$$

where $R_s = \frac{\mathcal{M}_{\text{fu}}^{\text{sclip}}(\boldsymbol{y}_w|\boldsymbol{x})}{\mathcal{M}_{\text{fu}}^{\text{sclip}}(\boldsymbol{y}_l|\boldsymbol{x})}$ and $R_p = \frac{\mathcal{M}^p(\boldsymbol{y}_w|\boldsymbol{x})}{\mathcal{M}^p(\boldsymbol{y}_l|\boldsymbol{x})}$. Analogous to DPO, the InfiFPO gradient increases the likelihood of preferred responses $\boldsymbol{y}_w$ while decreasing that of dispreferred responses $\boldsymbol{y}_l$. However, the critical distinction lies in the preference disparity coefficient, which weights training samples based on the divergence between the source and pivot models' preference assessments. This coefficient becomes larger when source models exhibit a stronger preference between $\boldsymbol{y}_w$ and $\boldsymbol{y}_l$ than the pivot model does. Consequently, samples where the source models strongly differentiate between responses but the pivot model does not yet reflect this distinction receive greater optimization emphasis. This adaptive weighting mechanism efficiently transfers preference knowledge from source models to the pivot model. The full derivation of the gradient is in Appendix A.2.

## 4 Experiments

We evaluated InfiFPO using Phi-4 as the pivot model and five mainstream open-source LLMs (9B∼24B parameters) as source models across 11 diverse benchmarks. Results show InfiFPO outperforms existing fusion and preference optimization methods.

### 4.1 Setup

**Model.** We use Phi-4 [7] as the pivot model. The source models consist of three general-purpose models (Qwen2.5-14B-Instruct [10], Mistral-Small-24B-Instruct[4], and Gemma-3-12B-Instruct [11]) and two domain-specific models (Qwen2.5-Coder-14B-Instruct [12] and Qwen2.5-Math-7B-Instruct [13]). By including these domain-specific models as additional source models, we can also investigate whether specialized expertise can enhance the overall performance of general models[5].

**Dataset.** We constructed a new training dataset comprising 150k examples across mathematics, coding, and general tasks. Data sources include Infinity-Instruct [14], NuminaMath-1.5 [15], and KodCode-V1-SFT [16], with detailed statistics provided

| Types | General Data | Math Data | Code Data |
|---|---|---|---|
| Dataset | Infinity-Instruct | NuminaMath-1.5 | KodCode-V1-SFT |
| Original Size | 1.4M | 1.4M | 268k |
| Sample Size | 60K | 45K | 45K |

Table 1: Dataset Statistics.

in Table 1. Since the original dataset may contain responses from outdated LLMs, potentially less capable than our pivot model, we retained only the prompts to build a new preference dataset. For each prompt, we generated responses using multiple models: 4 from each source model and 8 from the pivot model, all with a sampling temperature of 0.8. We then employed a reward model[6] [17] to evaluate these responses, selecting the highest-scored response as $\boldsymbol{y}_w$ and the lowest-scored as $\boldsymbol{y}_l$.

**Training Detail.** Our training process involved two stages with a batch size of 128 and a maximum sequence length of 4,096 tokens, using 16 NVIDIA A800-80GB GPUs. We implemented a cosine learning rate schedule with a 10% warmup ratio. In the first stage, we performed SFT on half of our dataset with $\boldsymbol{y}_w$ for 3 epochs, using a learning rate of 1e-6 to build the SFT model. This model then served as the foundation for the second stage, where we conducted Preference Optimization on the remaining half of the data for a single epoch, with a learning rate of 1e-7 and $\beta = 2.5$.

**Evaluation.** We conduct a comprehensive evaluation across 11 diverse benchmarks to assess the model's capabilities. Our evaluation spans five critical dimensions: (1) General reasoning (BBH [18], ARC-C [19], MMLU [20]), (2) Math (GSM8K [21], MATH [22], TheoremQA [23]), (3) Code (MBPP [24], HumanEval [25]), (4) Text reasoning (DROP [26], HellaSwag [27]), and (5) Instruction following (IFEval [28]). This multifaceted evaluation strategy enables us to systematically analyze the model's strengths and limitations across a spectrum of tasks requiring different cognitive abilities. More evaluation details are listed in Appendix B.

**Baseline.** We compare InfiFPO with three categories of baseness, including pivot &source model, model fusion, and preference optimization methods. For model fusion methods, we include FuseLLM [1], FuseChat [2], and InfiFusion [4]. Limited by the complex vocabulary alignment

---

[4]https://huggingface.co/mistralai/Mistral-Small-24B-Instruct-2501

[5]As shown in Table 2, domain-specific models Qwen-Coder and Qwen-Math perform poorly outside their specialized domains. To prevent these limited capabilities from being fused into the pivot model, we selectively fused Qwen-Coder only on code data and Qwen-Math only on mathematical data.

[6]https://huggingface.co/Skywork/Skywork-Reward-Gemma-2-27B

| Models | Math | | | Code | | General Reasoning | | | InstFol | Text Reasoning | | Avg | Model Size | GPU Hours |
|---|---|---|---|---|---|---|---|---|---|---|---|---|---|---|
| | GSM8K | MATH | ThmQA | MBPP | HEval | BBH | ARC | MMLU | IFEval | DROP | HS | | | |
| **Pivot Model** | | | | | | | | | | | | | | |
| Phi-4 | 87.41 | 80.04 | 51.12 | 75.40 | 83.54 | 68.84 | 93.90 | 85.62 | 77.34 | 88.67 | 87.62 | 79.95 | 14B | ∼1.0M |
| **Source Models** | | | | | | | | | | | | | | |
| Qwen2.5-Instruct | 91.13 | 78.16 | 47.25 | 81.70 | 83.54 | 77.59 | 92.20 | 80.22 | 85.01 | 85.56 | 88.28 | 80.97 | 14B | ∼1.8M |
| Mistral-Small | 92.42 | 69.84 | 48.50 | 68.80 | 84.15 | 81.59 | 91.86 | 81.69 | 82.25 | 86.52 | 91.84 | 79.95 | 24B | ∼1.6M |
| Qwen2.5-Coder | 89.16 | 74.18 | 38.88 | 85.40 | 90.90 | 75.40 | 89.49 | 75.08 | 74.70 | 84.34 | 79.83 | 77.94 | 14B | ∼1.8M |
| Gemma-3-Instruct | 93.71 | 82.90 | 49.62 | 72.60 | 82.32 | 85.70 | 71.19 | 77.61 | 90.77 | 86.43 | 83.34 | 79.65 | 12B | - |
| Qwen2.5-Math | 92.27 | 81.70 | 20.25 | 1.40 | 46.34 | 33.12 | 65.76 | 40.20 | 35.49 | 81.96 | 25.57 | 47.64 | 7B | ∼0.5M |
| **Model Fusion Methods** | | | | | | | | | | | | | | |
| FuseLLM* | 90.24 | 80.25 | 53.52 | 79.28 | 84.00 | 77.62 | 92.08 | 83.92 | 78.56 | 88.74 | 87.81 | 81.46 | 14B | 225 |
| FuseChat* | 91.21 | 77.52 | 51.88 | 81.80 | 84.15 | 83.37 | 93.56 | 84.23 | 78.90 | 89.23 | 87.42 | 82.12 | 14B | 650 |
| InfiFusion* | 90.07 | 80.94 | 55.62 | 81.80 | 83.54 | 80.94 | 94.24 | 85.81 | 76.02 | 89.27 | 87.91 | 82.38 | 14B | 160 |
| **Preference Optimization Methods** | | | | | | | | | | | | | | |
| SFT | 88.70 | 79.58 | 55.12 | 78.20 | 86.59 | 74.66 | 93.56 | 84.36 | 80.06 | 88.72 | 87.75 | 81.57 | 14B | 15 |
| SFT-DPO | 89.76 | 80.02 | 57.88 | 82.50 | 84.76 | 77.86 | 94.58 | 84.27 | 81.89 | 88.56 | 87.31 | 82.67 | 14B | 50 |
| SFT-IPO | 90.45 | 80.18 | 55.25 | 82.50 | 85.37 | 77.13 | 94.24 | 84.08 | 80.94 | 88.67 | 87.36 | 82.38 | 14B | 50 |
| SFT-WRPO | 89.92 | 80.02 | 57.88 | 83.10 | 86.59 | 78.18 | 94.24 | 83.98 | 81.18 | 88.41 | 87.30 | 82.80 | 14B | 57 |
| **InfiFPO** | | | | | | | | | | | | | | |
| InfiFPO* | 89.92 | 79.88 | 57.00 | 82.00 | 85.98 | 81.26 | 94.24 | 83.33 | 80.46 | 88.68 | 87.36 | 82.74 | 14B | 55 |
| InfiFPO | 90.07 | 80.10 | 57.25 | 82.50 | 87.80 | 82.02 | 94.24 | 84.27 | 82.25 | 88.83 | 87.29 | 83.33 | 14B | 58 |

Table 2: Main Results. * indicates that this method only uses Qwen2.5-Instruct, Qwen2.5-Coder, and Mistral-Small as source models. Abbreviations: InstFol (Instruction Following), ThmQA (TheoremQA), HEval (HumanEval), HS (HellaSwag). We use the packing technique for SFT, where multiple short examples are packed into a single input sequence to increase training efficiency.

and distribution merging process, we only include Qwen2.5-Instruct, Qwen2.5-Coder, and Mistral-Small as source models. For fair comparison, we select the same source models to implement InfiFPO (marked with asterisks). For preference optimization methods, we include DPO [8], IPO [29], and WRPO [6]. All these preference optimization methods adopt the same two-stage training approach as InfiFPO, i.e., first using half of the data for SFT, then using the remaining half for preference optimization.

## 4.2 Main Results

Table 2 presents comprehensive evaluation results across 11 benchmarks for different model types. From these results, we can draw several findings about InfiFPO.

**InfiFPO effectively integrates the capabilities of the source models.** InfiFPO significantly improved the pivot model's average performance across 11 benchmarks from 79.95 to 83.33. This integration is particularly remarkable as InfiFPO manages to inherit specialized strengths from diverse source models, such as mathematical reasoning from Qwen2.5-Math and code generation from Qwen2.5-Coder, while avoiding their respective weaknesses. For instance, while Qwen2.5-Math excels on GSM8K and MATH (92.27 and 81.70) but performs poorly on other tasks, and Qwen2.5-Coder achieves top scores on MBPP and HumanEval (85.40 and 90.90) but underperforms on theorem questions, InfiFPO maintains balanced high performance across these diverse task categories.

**InfiFPO outperforms model fusion baselines in both efficiency and effectiveness.** Compared to InfiFusion*, the best-performing baseline on average, InfiFPO* shows an average improvement of 0.36 across 11 benchmarks. The improvements are particularly significant for instruction-following and code tasks, with a 4.4 improvement on IFEval and a 2.4 improvement on HumanEval. More importantly, InfiFPO* requires only 34% of the GPU hours compared to InfiFusion* (55 vs 160). This is thanks to our implicit model fusion objective, which replaces token-level KL with sequence-level KL. This strategy preserves probability information while avoiding complex vocabulary alignment processes, making InfiFPO more efficient and scalable.

**InfiFPO consistently outperforms preference optimization baselines.** After training on half of the data's preferred responses $y_w$, SFT improved by 1.62 on average compared to the original model. Then, using the remaining half of the data for preference optimization, InfiFPO outperformed all preference optimization baselines. Compared to SFT-WRPO (the best-performing baseline), InfiFPO showed an average improvement of 0.53 across 11 benchmarks while using about the same

amount of GPU hours (58 vs 57). This demonstrates that our InfiFPO can better incorporate source model knowledge to enhance performance without significantly increasing training time compared to existing preference optimization baselines.

## 4.3 Analysis

We conduct two analyses: 1) examining the effectiveness of InfiFPO when combined with other preference optimization objectives; and 2) evaluating how different fusion strategies influence InfiFPO's performance. Due to space limitations, we only report three metrics: **Math** represents the average results from GSM8K, MATH, and ThmQA; **Code** represents the average of MBPP and HEval; and **All** represents the average results across all 11 benchmarks.

**Adaptation to Other Preference Optimization Objectives.** InfiFPO is fundamentally an improvement on the reference model, making it applicable not only to DPO but also to other DPO variants. To verify the versatility of InfiFPO, we integrated it with IPO and WRPO, creating InfiFPO$_{IPO}$ and InfiFPO$_{WRPO}$ respectively. Table 3 demonstrates InfiFPO's versatility as a general enhancement framework that can be effectively integrated with various preference optimization objectives beyond standard DPO. When integrated with WRPO,

| Method | Math | Code | All |
|---|---|---|---|
| WRPO | 75.94 ↓0.0 | 84.84 ↓0.0 | 82.80 ↓0.0 |
| InfiFPO$_{WRPO}$ | 76.31 ↑0.4 | 85.20 ↑0.2 | 83.32 ↑0.5 |
| IPO | 75.29 ↓0.0 | 83.93 ↓0.0 | 82.38 ↓0.0 |
| InfiFPO$_{IPO}$ | 76.37 ↑1.1 | 84.54 ↑0.5 | 83.15 ↑0.8 |

Table 3: Results of InfiFPO combined with different preference optimization objectives.

InfiFPO delivers modest but consistent gains in mathematical reasoning (+0.4), code generation (+0.2), and overall performance (+0.5). The improvements are more pronounced when combining InfiFPO with IPO, yielding notable enhancements in Math (+1.1) and Code (+0.5) benchmarks, with an overall improvement of +0.8 across all tasks. These results empirically validate that InfiFPO's fusion-based approach provides complementary benefits to existing preference optimization techniques by leveraging diverse source models while maintaining the pivot model's optimization objective. The loss functions for InfiFPO$_{IPO}$ and InfiFPO$_{WRPO}$ can be found in Appendix C.

**Different Fusion Strategies.** In §3.2, we proposed the max-marginal fusion strategy to extract unique information from source models. Additionally, we explored two other fusion strategies: an average-based strategy and a confidence-based strategy. For the average-based strategy, we treat all source models equally, assigning each a weight of $1/N$. For the confidence-based strategy, we weight source models according to their confidence in the response, measured by sequence probability, meaning models with higher confidence receive greater weights. Figure 2 illustrates the performance of InfiFPO under different fusion strategies. We observe

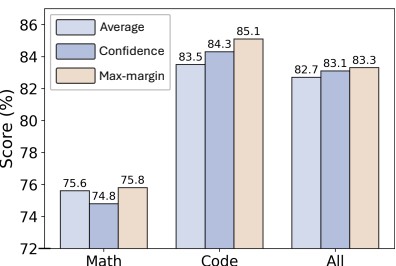

Figure 2: Different fusion strategies.

that the Confidence-based strategy slightly outperforms the Average-based strategy, likely because it emphasizes high-confidence models during fusion, thus acquiring better information. The Max-marginal strategy consistently outperforms both the Average-based and Confidence-based strategies, as it focuses on the most distinctive models, thereby learning more unique and complementary information. Detailed fusion strategies can be found in Appendix D.

## 4.4 Ablation Study

To investigate the effects of different components in InfiFPO, we conducted ablation experiments in three parts: Length Normalization & Probability clipping, Number of Source Models.

**Length Normalization and Probability Clipping.** Table 4 demonstrates the critical importance of both techniques. Without these optimizations, InfiFPO actually underperforms the baseline Phi4 model. Length Normalization alone provides substantial improvements across all metrics (+2.4 in Math, +4.5 in Code, +2.9 in All), addressing the length bias issue. Similarly, Probability Clipping

| Method | LN | PC | Math | Code | All |
|---|---|---|---|---|---|
| Phi4 | | | 72.85 ↓0.0 | 79.47 ↓0.0 | 79.95 ↓0.0 |
| InfiFPO | | | 72.72 ↓0.1 | 78.42 ↓1.0 | 79.51 ↓0.4 |
| | ✓ | | 75.06 ↑2.4 | 83.99 ↑4.5 | 82.86 ↑2.9 |
| | | ✓ | 74.39 ↑1.3 | 81.32 ↑1.9 | 81.41 ↑1.5 |
| | ✓ | ✓ | **75.80 ↑3.1** | **85.15 ↑5.6** | **83.33 ↑3.3** |

Table 4: Ablation Study on Length Normalization (LN) and Probability Clipping (PC).

yields notable gains (+1.3 in Math, +1.9 in Code,
+1.5 in All) by preventing source model degradation. The combination of both techniques delivers
the strongest performance boost, with significant improvements across all benchmarks (+3.1 in Math,
+5.6 in Code, +3.3 in All), confirming their complementary nature in enhancing model capabilities.

**Number of Source Models.** We conducted experiments on
InfiFPO with an increasing number of source models, adding
different models one by one for fusion. The experiment
started with a single source model, Qwen2.5-Instruct, then
gradually added Mistral-Small, Qwen2.5-Coder, Gemma-
3-Instruct, and Qwen2.5-Math. Table 5 shows clear per-
formance improvements as the number of source models
increases. With each additional model, we observe consis-
tent gains across all metrics, particularly in Code tasks (+3.2
points with 5 models). The diminishing returns after 4-5

| Num | Math | Code | All |
|---|---|---|---|
| 1 | 74.91 ↓0.0 | 81.91 ↓0.0 | 81.54 ↓0.0 |
| 2 | 75.53 ↑0.6 | 82.49 ↑0.5 | 82.20 ↑0.6 |
| 3 | 75.60 ↑0.6 | 83.99 ↑2.0 | 82.74 ↑1.2 |
| 4 | 75.25 ↑0.3 | **85.21 ↑3.3** | 83.14 ↑1.6 |
| 5 | **75.80 ↑0.8** | 85.15 ↑3.2 | **83.33 ↑1.7** |

Table 5: Ablation Study on Number of Source Models.

models suggest a balance between performance benefits and computational resources when selecting
source models for fusion.

## 5 Related Work

**Preference Optimization.** Aligning LLMs with human preferences often relies on RLHF [30, 31],
where a reward model is trained from human comparisons and used in PPO to optimize the policy.
While effective, RLHF is resource-intensive and complex to converge. To simplify preference
alignment, DPO [8] reformulates the objective as offline learning from preference pairs, removing the
need for a reward model and RL. Azar et al. [29] unify RLHF and DPO under a general preference
optimization framework, and propose IPO, a variant that avoids overfitting by using the identity
transformation, offering more stable learning in low-data or biased settings. Gu et al. [32] further
propose the PAD framework, which models preference knowledge as a probability distribution over
responses, providing more nuanced supervisory signals for preferences distilling.

**Model Fusion.** Integrating models aims to combine multiple LLMs into a single model that inherits
their respective strengths. While earlier methods like model merging [33–38] require architectural
compatibility, fusion techniques relax such constraints, making them more suitable for heterogeneous
models. Another line of work, knowledge distillation (KD) [39–41], offers an alternative approach
for transferring capabilities across models without requiring structural homogeneity. KD transfers
the generalization ability of larger models to smaller ones by leveraging "soft targets" the output
probability distributions (logits) of the teacher model enabling efficient deployment while preserving
performance. However, traditional KD typically assumes that the teacher model is larger than the
student and fully covers the target capability space. Moreover, most prior work has focused on the
single teacher setting [42, 43], limiting its flexibility in multi-teacher scenarios.

FuseLLM [2] introduced LLM fusion, showing gains in reasoning and code tasks. Later works
extended this idea across domains [44], but struggled with structural mismatches. To address this,
pairwise fusion [2] sequentially integrates models into the pivot model, and InfiFusion [4] improved
it via adaptive merging and unified output aggregation. The most explicit fusion approaches operate
at the token level and face vocabulary alignment issues. WRPO [6] mitigates this by shifting fusion
to the sequence level using reinforcement learning, enabling implicit fusion without vocabulary
conflicts. However, WRPO has two major limitations: it discards probabilistic outputs using only
response-level supervision and focuses solely on preferred responses while ignoring contrastive
signals from dispreferred ones, resulting in only partial leverage of source model capabilities.

## 6 Conclusion

We introduce **InfiFPO**, a novel framework that enables model fusion during the preference alignment
phase by replacing the reference model in DPO with a fused sequence-level distribution over multiple
source models. Unlike prior work, InfiFPO leverages full-sequence probabilities rather than token-
level outputs, thereby avoiding vocabulary alignment issues across heterogeneous models while
preserving rich preference signals. To make this optimization practical and efficient, we derive an
offline training objective from a sequence-KL constrained RLHF formulation, termed FuseRLHF.

We further enhance stability through three key strategies: length normalization to mitigate sequence bias, probability clipping to suppress noisy gradients, and max-margin fusion to prioritize diverse and informative sources. Experiments across 11 benchmarks demonstrate that InfiFPO consistently outperforms strong baselines in both model capability and alignment quality. Our results highlight that preference optimization not only accommodates but can significantly benefit from principled model fusion, offering a robust and scalable path to integrating diverse LLMs.

## Limitation

Despite InfiFPO demonstrating significant empirical performance, it still relies on existing preference optimization methods such as DPO. More rigorous theoretical analysis is needed to better understand InfiFPO's fusion mechanisms and further strengthen its theoretical foundation. Additionally, due to computational resource constraints, we only selected five mainstream open-source LLMs as source models for our experiments, which cannot represent the SOTA performance of current advanced LLMs. Experiments with larger-scale models and datasets remain unexplored.

## Funding Transparency Statement

**Funding in direct support of this work.** This research was supported by the team of Prof. Hongxia Yang at The Hong Kong Polytechnic University. GPU resources were donated by InfiX.ai.

**Competing Interests.** The authors declare no competing interests.

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

## A Mathematical Derivations

### A.1 Deriving the FPO Objective

In this appendix, we will derive Eq. (7). Based on Eq. (14), we optimize the following objective:

$$\arg\max_{\mathcal{M}^p} \mathbb{E}_{\boldsymbol{x}\sim\mathcal{D},\boldsymbol{y}\sim\mathcal{M}^p(\boldsymbol{y}|\boldsymbol{x})}[\boldsymbol{r}(\boldsymbol{x},\boldsymbol{y})] - \beta\sum_i^N \gamma_i\bigg(\log\mathcal{M}^p(\boldsymbol{y}|\boldsymbol{x}) - \log\mathcal{M}_i^s(\boldsymbol{y}|\boldsymbol{x})\bigg), \quad (14)$$

then we have:

$$\arg\max_{\mathcal{M}^p} \mathbb{E}_{\boldsymbol{x}\sim\mathcal{D},\boldsymbol{y}\sim\mathcal{M}^p(\boldsymbol{y}|\boldsymbol{x})}[\boldsymbol{r}(\boldsymbol{x},\boldsymbol{y})] - \beta\sum_i^N \gamma_i\mathbb{D}_{\mathrm{SKL}}\left[\mathcal{M}^p(\boldsymbol{y}|\boldsymbol{x})\|\mathcal{M}_i^s(\boldsymbol{y}|\boldsymbol{x})\right]$$

$$= \arg\max_{\mathcal{M}^p} \mathbb{E}_{\boldsymbol{x}\sim\mathcal{D}}\mathbb{E}_{\boldsymbol{y}\sim\mathcal{M}^p(\boldsymbol{y}|\boldsymbol{x})}\left[\boldsymbol{r}(\boldsymbol{x},\boldsymbol{y}) - \beta\log\frac{\mathcal{M}^p(\boldsymbol{y}|\boldsymbol{x})}{\mathcal{M}_{\mathrm{fu}}^s(\boldsymbol{y}|\boldsymbol{x})}\right]$$

$$= \arg\min_{\mathcal{M}^p} \mathbb{E}_{\boldsymbol{x}\sim\mathcal{D}}\mathbb{E}_{\boldsymbol{y}\sim\mathcal{M}^p(\boldsymbol{y}|\boldsymbol{x})}\left[\log\frac{\mathcal{M}^p(\boldsymbol{y}|\boldsymbol{x})}{\mathcal{M}_{\mathrm{fu}}^s(\boldsymbol{y}|\boldsymbol{x})} - \frac{1}{\beta}\boldsymbol{r}(\boldsymbol{x},\boldsymbol{y})\right]$$

$$= \arg\min_{\mathcal{M}^p} \mathbb{E}_{\boldsymbol{x}\sim\mathcal{D}}\mathbb{E}_{\boldsymbol{y}\sim\mathcal{M}^p(\boldsymbol{y}|\boldsymbol{x})}\left[\log\frac{\mathcal{M}^p(\boldsymbol{y}|\boldsymbol{x})}{\frac{1}{Z(\boldsymbol{x})}\mathcal{M}_{\mathrm{fu}}^s(\boldsymbol{y}|\boldsymbol{x})\exp(\frac{1}{\beta}\boldsymbol{r}(\boldsymbol{x},\boldsymbol{y}))} - \log Z(\boldsymbol{x})\right] \quad (15)$$

where the fused source model $\mathcal{M}_{\mathrm{fu}}^s(\boldsymbol{y}|\boldsymbol{x}) = \prod_{i=1}^{N}\left(\mathcal{M}_i^s(\boldsymbol{y}|\boldsymbol{x})\right)^{\gamma_i}$ and the partition function $Z(\boldsymbol{x}) = \sum_{\boldsymbol{y}} \mathcal{M}_{\mathrm{fu}}^s(\boldsymbol{y}|\boldsymbol{x})\exp\left(\frac{1}{\beta}\boldsymbol{r}(\boldsymbol{x},\boldsymbol{y})\right)$.

Note that the partition function is a function of only $\boldsymbol{x}$ and the fused source model $\mathcal{M}_{\mathrm{fu}}^s$, but does not depend on the pivot model $\mathcal{M}^p$. Hence we have the optimal solution $\mathcal{M}^{p*}$:

$$\mathcal{M}^{p*}(\boldsymbol{y}|\boldsymbol{x}) = \frac{1}{Z(x)}\mathcal{M}_{\mathrm{fu}}^s(\boldsymbol{y}|\boldsymbol{x})\exp(\frac{1}{\beta}\boldsymbol{r}(\boldsymbol{x},\boldsymbol{y})). \tag{16}$$

This completes the derivation.

### A.2 Deriving the Gradient of the InfiFPO Objective

In this section, we derive the gradient of the InfiFPO objective, where the parameters of the pivot model are denoted as $\theta$:

$$\nabla\theta\mathcal{L}_{\mathrm{InfiFPO}}(\mathcal{M}^p;\{\mathcal{M}_i^s\}_{i=1}^N) = -\nabla\theta\mathbb{E}_{(\boldsymbol{x},\boldsymbol{y}_w,\boldsymbol{y}_l)\sim\mathcal{D}^P}\left[\log\sigma\left(\beta\overline{\log}\frac{\mathcal{M}^P(\boldsymbol{y}_w|\boldsymbol{x})}{\mathcal{M}_{\mathrm{fu}}^{\mathrm{sclip}}(\boldsymbol{y}_w|\boldsymbol{x})} - \beta\overline{\log}\frac{\mathcal{M}^P(\boldsymbol{y}_l|\boldsymbol{x})}{\mathcal{M}_{\mathrm{fu}}^{\mathrm{sclip}}(\boldsymbol{y}_l|\boldsymbol{x})}\right)\right],$$
$$\tag{17}$$

where $\mathcal{M}_{\mathrm{fu}}^{\mathrm{sclip}}(\boldsymbol{y}|\boldsymbol{x}) = \prod_{i=1}^{N}\left[\mathrm{Clip}(\mathcal{M}_i^s(\boldsymbol{y}|\boldsymbol{x}))\right]^{\gamma_i}$.

We can rewrite Eq. (17) as

$$\nabla\theta\mathcal{L}_{\mathrm{InfiFPO}}(\mathcal{M}^p;\{\mathcal{M}_i^s\}_{i=1}^N) = -\nabla\theta\mathbb{E}_{(\boldsymbol{x},\boldsymbol{y}_w,\boldsymbol{y}_l)\sim\mathcal{D}^P}\left[\log\sigma\left(\frac{\sigma'(u)}{\sigma(u)}\nabla\theta(u)\right)\right], \tag{18}$$

where $u = \beta\overline{\log}\frac{\mathcal{M}^P(\boldsymbol{y}_w|\boldsymbol{x})}{\mathcal{M}_{\mathrm{fu}}^{\mathrm{sclip}}(\boldsymbol{y}_w|\boldsymbol{x})} - \beta\overline{\log}\frac{\mathcal{M}^P(\boldsymbol{y}_l|\boldsymbol{x})}{\mathcal{M}_{\mathrm{fu}}^{\mathrm{sclip}}(\boldsymbol{y}_l|\boldsymbol{x})}$.

With the properties of sigmoid function $\sigma'(x) = \sigma(x)(1-\sigma(x))$ and $\sigma(x) = 1 - \sigma(-x)$, we can get the gradient:

$$\nabla\theta\mathcal{L}_{\mathrm{InfiFPO}} =$$

$$-\beta\mathbb{E}_{(\boldsymbol{x},\boldsymbol{y}_w,\boldsymbol{y}_l)\sim\mathcal{D}^P}\left[\sigma\left(\beta\overline{\log}\frac{\mathcal{M}^P(\boldsymbol{y}_l|\boldsymbol{x})}{\mathcal{M}_{\mathrm{fu}}^{\mathrm{sclip}}(\boldsymbol{y}_l|\boldsymbol{x})} - \beta\overline{\log}\frac{\mathcal{M}^P(\boldsymbol{y}_w|\boldsymbol{x})}{\mathcal{M}_{\mathrm{fu}}^{\mathrm{sclip}}(\boldsymbol{y}_w|\boldsymbol{x})}\right)\left[\nabla\theta\log\mathcal{M}^P(\boldsymbol{y}_w|\boldsymbol{x}) - \nabla\theta\log\mathcal{M}^P(\boldsymbol{y}_l|\boldsymbol{x})\right]\right].$$

After rewriting the first term, we obtain the final form of the gradient in § 3.3.

## B Evaluation Setup

We adopt OpenCompass [45] and EvalPlus [46] to conduct evaluation on 11 benchmark datasets. We revise the prompts of certain datasets within OpenCompass to ensure more reliable answer extraction via its regex-based matching mechanism. The detailed prompts are listed in Table 6.

For TheremQA and HumanEval, we follow the default evaluation settings without modifying the prompts. For MBPP, we employ EvalPlus [46] for rigourous evaluation of LLM-synthesized code.

## C Adaptation to Other Preference Optimization Objectives

The original objective of WRPO is

$$\mathcal{L}_{\mathrm{WRPO}}(\mathcal{M}_\theta;\mathcal{M}_{\mathrm{ref}}) = -\mathbb{E}_{(\boldsymbol{x},\boldsymbol{y}_{w_s},\boldsymbol{y}_{w_p},\boldsymbol{y}_l)\sim\mathcal{D}}$$
$$\left[\log\sigma\left(\alpha\cdot\beta\log\frac{\mathcal{M}_\theta(\boldsymbol{y}_{w_s}|x)}{\mathcal{M}_{\mathrm{ref}}(\boldsymbol{y}_{w_s}|x)} + (1-\alpha)\cdot\beta\log\frac{\mathcal{M}_\theta(\boldsymbol{y}_{w_p}|x)}{\mathcal{M}_{\mathrm{ref}}(\boldsymbol{y}_{w_p}|x)} - \beta\log\frac{\mathcal{M}_\theta(\boldsymbol{y}_l|x)}{\mathcal{M}_{\mathrm{ref}}(\boldsymbol{y}_l|x)}\right)\right], \tag{19}$$

where $\boldsymbol{y}_{w_s}$ is the preferred response generated by source models, and $\boldsymbol{y}_{w_p}$ is the dispreferred response generated by the pivot model. $\alpha$ represents the fusion coefficient that dynamically balances the reward of $\boldsymbol{y}_{w_s}$ and $\boldsymbol{y}_{w_p}$. After adapting InfiFPO to this objective, we can obtain

$$\mathcal{L}_{\mathrm{InfiFPO\text{-}WRPO}}(\mathcal{M}^p;\{\mathcal{M}_i^s\}_{i=1}^N) = -\mathbb{E}_{(\boldsymbol{x},\boldsymbol{y}_{w_s},\boldsymbol{y}_{w_p},\boldsymbol{y}_l)\sim\mathcal{D}}$$
$$\left[\overline{\log}\sigma\left(\alpha\cdot\beta\overline{\log}\frac{\mathcal{M}_\theta(\boldsymbol{y}_{w_s}|x)}{\mathcal{M}_{\mathrm{fu}}^{\mathrm{sclip}}(\boldsymbol{y}_{w_s}|x)} + (1-\alpha)\cdot\beta\overline{\log}\frac{\mathcal{M}_\theta(\boldsymbol{y}_{w_p}|x)}{\mathcal{M}_{\mathrm{fu}}^{\mathrm{sclip}}(\boldsymbol{y}_{w_p}|x)} - \beta\overline{\log}\frac{\mathcal{M}_\theta(\boldsymbol{y}_l|x)}{\mathcal{M}_{\mathrm{fu}}^{\mathrm{sclip}}(\boldsymbol{y}_l|x)}\right)\right]. \tag{20}$$

Table 6: Prompt format used for different datasets.

| Dataset | Prompt Format |
|---------|---------------|
| IFEval | `{prompt}\nPlease directly give the correct answer:` |
| ARC-C | `Question: {question}\nA . {textA}\nB . {textB}\nC . {textC}\nD . {textD}\nDirectly give me the correct answer option, and then explain:` |
| Hellaswag | `{ctx}\nQuestion : Which ending makes the most sense?\nDirectly give me the correct choice, you can further explain it or not.\nA . {A}\nB . {B}\nC . {C}\nD . {D}\nYou may choose from 'A', 'B', 'C', 'D'.\nAnswer :` |
| BBH | `Follow the given examples and answer the question.\n {_hint}\nQ : {{input}}\nA : Let's think step by step.` |
| DROP | `You will be asked to read a passage and answer a question. Some examples of passages and Q\&A are provided below.\n {drop_examples}\n \n ## Your Task\n --\n {prompt}\nThink step by step, then write a line of the form "Answer: \$ANSWER" at the end of your response.` |
| MMLU | `{_hint}\nQuestion : {{input}}\nA . {{A}}\nB . {{B}}\nC . {{C}}\nD . {{D}}\n \nFor simple problems:\nDirectly provide the answer with minimal explanation.\n \nFor complex problems:\nUse this step-by-step format:\n #### Step 1: [Concise description]\n [Brief explanation]\n #### Step 2: [Concise description]\n [Brief explanation]\n \nRegardless of the approach, always conclude with:\nThe answer is [the\_answer\_letter].\nwhere the [the\_answer\_letter] is one of A, B, C or D.\n \nLet 's think step by step.` |
| GSM8K | `{question}\nPlease reason step by step, and put your final answer within \\boxed\{\}.` |
| MATH | `{problem}\nPlease reason step by step, and put your final answer within \\boxed\{\}.` |

The original objective of the IPO is

$$\mathcal{L}_{\text{IPO}}(\mathcal{M}_\theta; \mathcal{M}_{\text{ref}}) = -\mathbb{E}_{(\boldsymbol{x},\boldsymbol{y}_w,\boldsymbol{y}_l)\sim\mathcal{D}^p} \left[ \left( \beta\overline{\log}\frac{\mathcal{M}_\theta(\boldsymbol{y}_w|\boldsymbol{x})}{\mathcal{M}_{\text{ref}}(\boldsymbol{y}_w|\boldsymbol{x})} - \beta\overline{\log}\frac{\mathcal{M}_\theta(\boldsymbol{y}_l|\boldsymbol{x})}{\mathcal{M}_{\text{ref}}(\boldsymbol{y}_l|\boldsymbol{x})} \right) - \frac{1}{2\beta} \right]^2 . \quad (21)$$

Since IPO already implements length normalization[7], the main changes when adapting to InfiFPO are in the reference model part.

$$\mathcal{L}_{\text{InfiFPO-IPO}}(\mathcal{M}^p; \{\mathcal{M}_i^s\}_{i=1}^N) = -\mathbb{E}_{(\boldsymbol{x},\boldsymbol{y}_w,\boldsymbol{y}_l)\sim\mathcal{D}^p} \left[ \left( \overline{\log}\frac{\mathcal{M}^p(\boldsymbol{y}_w|\boldsymbol{x})}{\mathcal{M}_{\text{fu}}^{\text{sclip}}(\boldsymbol{y}_w|\boldsymbol{x})} - \overline{\log}\frac{\mathcal{M}^p(\boldsymbol{y}_l|\boldsymbol{x})}{\mathcal{M}_{\text{fu}}^{\text{sclip}}(\boldsymbol{y}_l|\boldsymbol{x})} \right) - \frac{1}{2\beta} \right]^2$$

$$(22)$$

# D Different Fusion Strategies

When we focus on the fused source model $\mathcal{M}_{\text{fu}}^s$, which is the weighted geometric mean of all source model sequence probabilities, we can find that it originates from the model fusion objective. By changing its weights, we can adopt different fusion strategies. In addition to the Max-margin fusion strategy mentioned in § 3.2, we design two following strategies. Their results can be viewed in § 4.3.

**Average-based Fusion.** A common fusion strategy is to treat $N$ source models equally, adopting the same weight of $\frac{1}{N}$, thus the fusion model becomes

$$\mathcal{M}_{\text{fu}}^s = \prod_{i=1}^N (\mathcal{M}_i^s(\boldsymbol{y}|\boldsymbol{x}))^{\frac{1}{N}} \quad (23)$$

**Confidence-based Fusion.** To dynamically weight the source models, we introduce a confidence-based strategy and define the confidence of each source model $\mathcal{M}_i^s$ as the inverse of its negative log-likelihood:

$$\text{Confidence}_i(\boldsymbol{y}|\boldsymbol{x}) = \frac{1}{-\log \mathcal{M}_i^s(\boldsymbol{y}|\boldsymbol{x}) + \epsilon}, \quad (24)$$

---

[7]https://huggingface.co/blog/pref-tuning

where $\epsilon$ is a small constant ensuring numerical stability [47]. This design follows the intuition that $-\log \mathcal{M}_i^s(\boldsymbol{y}|\boldsymbol{x})$ quantifies the information content of a prediction, where smaller values indicate higher confidence. Taking its inverse reflects the common practice of inverse-uncertainty weighting [48], assigning larger weights to more confident models. This fusion mechanism emphasizes source models with higher confidence, allowing $\mathcal{M}_{\text{fu}}^s$ to adaptively inherit more reliable information from the sources. The corresponding normalized weight for $\mathcal{M}_i^s$ is given by SoftMax:

$$w_i(\boldsymbol{y}|\boldsymbol{x}) = \frac{\text{Confidence}_i(\boldsymbol{y}|\boldsymbol{x})}{\sum_{j=1}^{N} \text{Confidence}_j(\boldsymbol{y}|\boldsymbol{x})}. \tag{25}$$

Based on these confidence-derived weights, the fused model $\mathcal{M}_{\text{fu}}^s$ aggregates source models via:

$$\mathcal{M}_{\text{fu}}^s(\boldsymbol{y}|\boldsymbol{x}) = \prod_{i=1}^{N} \mathcal{M}_i^s(\boldsymbol{y}|\boldsymbol{x})^{w_i(\boldsymbol{y}|\boldsymbol{x})}. \tag{26}$$

# E   Supplementary Experiments and Analysis

## E.1   Additional Experimental Results

### E.1.1   Data Diversity Validation

To validate InfiFPO's robustness to data diversity, we conducted experiments on UltraFeedback [49], which contains approximately 60k preference pairs and has been widely adopted in preference optimization research. We treated the preferred (chosen) responses as labels for model fusion baselines.

We selected three general-purpose models for fusion: 1) Qwen2.5-14B-Instruct; 2) Mistral-Small; 3) Gemma-3-Instruct. We compared against InfiFusion and SFT-WRPO, which demonstrated strong performance in our main experiments. As shown in Table 7, InfiFPO remains effective on UltraFeedback and outperforms relevant baselines.

| Model/Method | Math | Code | All |
|---|---|---|---|
| Phi-4 | 72.86 | 79.47 | 79.95 |
| InfiFusion | 73.23 | 82.43 | 81.44 |
| SFT-WRPO | 73.15 | 82.21 | 81.58 |
| InfiFPO | **73.59** | **83.15** | **82.19** |

Table 7: Experimental results on UltraFeedback

However, the optimal performance on UltraFeedback (82.19) is notably lower than on our constructed dataset (83.33), validating the effectiveness of our data construction process.

### E.1.2   Strong Backbone Validation

We conducted experiments using Qwen2.5-14B-Instruct as the target (pivot) model, selecting three source models: 1) Phi-4; 2) Mistral-Small; 3) Gemma-3-Instruct. We used FuseChat, InfiFusion, and WRPO as baselines with identical experimental settings to our main experiments.

| Model / Method | Math | Code | All |
|---|---|---|---|
| Qwen-2.5-Instruct | 72.18 | 82.62 | 80.97 |
| FuseChat | 75.96 | 83.50 | 83.22 |
| InfiFusion | 76.09 | 83.38 | 83.16 |
| SFT-WRPO | 76.42 | 84.29 | 83.46 |
| InfiFPO | **76.46** | **85.41** | **84.01** |

Table 8: Experimental results using Qwen2.5-14B-Instruct as target model

Table 8 demonstrates that when using Qwen2.5-Instruct as the target model, our method still outperforms other baselines, proving its generalizability. In our main experiments, we chose Phi-4 over Qwen2.5-Instruct as the target model because Phi-4 is heterogeneous with all source models (different architectures, vocabularies, etc.), representing a more general and typical scenario for model fusion.

### E.1.3 Reward Model Sensitivity Analysis

To examine data sensitivity, we tested the impact of different reward models on method performance. We tested two reward models: Skywork-Reward-Llama-3.1-8B-v0.2 and ArmoRM-Llama3-8B-v0.1, both widely used in preference optimization.

| Model / Method | Math | Code | All |
|---|---|---|---|
| Skywork-Reward-Llama-3.1-8B-v0.2 | | | |
| Phi-4 | 72.86 | 79.47 | 79.95 |
| InfiFusion | 74.32 | 82.47 | 81.96 |
| SFT-WRPO | 74.28 | 83.44 | 81.93 |
| InfiFPO | **74.53** | **84.88** | **82.67** |
| ArmoRM-Llama3-8B-v0.1 | | | |
| Phi-4 | 72.86 | 79.47 | 79.95 |
| InfiFusion | 73.38 | 82.54 | 81.59 |
| SFT-WRPO | 73.07 | 83.04 | 81.62 |
| InfiFPO | **73.45** | **84.94** | **82.46** |

Table 9: Results using different reward models

We observe two phenomena from Table 9: (1) InfiFPO consistently outperforms baselines when using different reward models; (2) Different reward models yield different performance improvements. According to Reward Bench v2, Skywork-Reward-Gemma-2-27B-v0.2 significantly outperforms the other two reward models, indicating a positive correlation between reward model performance and final model performance.

### E.1.4 Sequence Alignment Ablation

We conducted ablation experiments to investigate the impact of sequence alignment on model fusion methods. We removed sequence alignment from all model fusion methods for fair comparison.

| Model / Method | Math | Code | All |
|---|---|---|---|
| Phi-4 | 72.86 | 79.47 | 79.95 |
| FuseChat | 73.54 | 82.97 | 82.12 |
| - without sequence alignment | 72.30 | 79.16 | 79.43 |
| InfiFusion | 75.54 | 82.67 | 82.38 |
| - without sequence alignment | 73.47 | 79.98 | 80.11 |
| InfiFPO | 75.60 | 83.99 | 82.74 |
| - without sequence alignment | **73.72** | **81.28** | **80.82** |

Table 10: Impact of sequence alignment on model fusion methods

Results in Table 10 show that even without sequence alignment, InfiFPO outperforms other model fusion baselines. Token-level fusion methods like FuseChat without sequence alignment even perform worse than the base Phi-4 model, indicating greater dependence on sequence alignment. Therefore, we consider addressing vocabulary conflicts essential for model fusion methods.

### E.1.5 Data Scaling Analysis

We tested different data volumes (25k/50k/75k/100k) during preference optimization to analyze performance scaling.

| Model / Method | 25k | 50k | 75k | 100k |
|---|---|---|---|---|
| Phi-4 | 79.95 | 79.95 | 79.95 | 79.95 |
| SFT-WRPO | 81.58 | 82.14 | 82.80 | 82.89 |
| InfiFPO | **82.55** | **82.99** | **83.33** | **83.56** |

Table 11: Performance scaling with data volume

Table 11 shows consistent performance improvements across different data volumes, with gains increasing with larger datasets, demonstrating the scalability of our approach.

### E.1.6    Small-scale Model Validation

We experimented with Qwen2.5-Instruct-1.5B/3B/7B as pivot models to validate our method's effectiveness across different model scales.

| Model / Method | Math | Code | All |
|---|---|---|---|
| Qwen2.5-1.5B-Instruct | | | |
| Base Model | 46.02 | 47.16 | 50.31 |
| InfiFusion | 49.59 | 50.49 | 53.11 |
| SFT-WRPO | 50.16 | 51.21 | 53.70 |
| InfiFPO | **50.70** | **52.75** | **54.61** |
| Qwen2.5-3B-Instruct | | | |
| Base Model | 55.88 | 62.26 | 64.07 |
| InfiFusion | 58.82 | 65.33 | 66.72 |
| SFT-WRPO | 59.31 | 65.63 | 67.01 |
| InfiFPO | **60.05** | **66.64** | **67.96** |
| Qwen2.5-7B-Instruct | | | |
| Base Model | 62.48 | 70.13 | 73.04 |
| InfiFusion | 65.09 | 72.24 | 75.35 |
| SFT-WRPO | 65.45 | 72.51 | 75.65 |
| InfiFPO | **66.14** | **73.31** | **76.24** |

Table 12: Results with different small-scale pivot models

Table 12 demonstrates that our method outperforms model fusion and preference optimization baselines across different pivot model sizes, confirming scalability and effectiveness.

### E.2    Technical Details and Method Analysis

### E.2.1    Technical Implementation Details

**Pre-computation Strategy:** In practice, we pre-compute log probabilities from target (pivot) and source models before training, which provides two key advantages: (1) Reduced training memory footprint by avoiding the need to load multiple source models during training; (2) Accelerated computation through efficient third-party inference libraries.

We employ vLLM for acceleration, requiring only approximately 8 GPU hours to compute log probabilities for 5 source models. This explains why our GPU hours increase by merely  10% compared to vanilla DPO when fusing multiple source models.

**Memory Efficiency:** InfiFPO offers significant memory efficiency advantages. It performs model fusion at the sequence level, requiring only sequence-level log probabilities (a few floating-point numbers per sequence). When the number of source models increases, training efficiency remains largely unaffected. In contrast, token-level fusion methods must load probability distributions over

the entire vocabulary ( 100k tokens per token postion), requiring 100k-scale floating-point numbers per token that scale linearly with source model count.

### E.2.2   In-depth Analysis of Method Principles

We provide a detailed discussion of how source models are selected during Probability Clipping (PC) and Max-Margin fusion strategies.

**Probability Clipping and Max-Margin Process:** PC aims to avoid interference from weak source models by clipping the corresponding log probabilities, ensuring that source models do not assign lower probabilities to preferred responses or higher probabilities to dispreferred responses than the pivot model. Max-Margin selects the source model with the most different preference knowledge from the pivot model as the reference model.

When using PC, Max-Margin selects the "smartest" source model, meaning the source model that assigns probabilities to preferred responses no lower than the pivot model, probabilities to dispreferred responses no higher than the pivot model, and has the largest margin between preferred and dispreferred responses.

**Example Illustration:** Given two responses, the probability values assigned by models are:

| Model type | Logps (preferred) | Logps (dispreferred) |
|---|---|---|
| Pivot model | -0.4 | -0.7 |
| Source model 1 | -0.6 | -0.5 |
| Source model 2 | -0.3 | -0.8 |

Table 13: Example of model probabilities before PC

After PC, the source models' log probabilities become:

| Model type | Logps (preferred) | Logps (dispreferred) | Selected |
|---|---|---|---|
| Pivot model | -0.4 | -0.7 | - |
| Source model 1 | -0.4 | -0.7 | - |
| Source model 2 | -0.3 | -0.8 | ✓ |

Table 14: Example of model probabilities after PC

Based on the Max-Margin strategy, source model 2 serves as the reference model. It is the "smartest" source model, having the highest log probabilities for preferred responses and the lowest log probabilities for dispreferred responses.

This strategy aligns with findings in FuseLLM, which adopts the MinCE strategy to select source models with the lowest cross-entropy for ground truth. Both approaches indicate that in model fusion, fusing knowledge from multiple source models may not be as effective as fusing knowledge from the "best" source model.

## F   Case Study

### F.1   Advantages in Code Generation

To evaluate the effectiveness of our proposed InfiFPO strategy, we conduct a detailed comparison with the baseline Supervised Fine-Tuning (SFT) model on the HumanEval benchmark.

Through case studies, we observe that InfiFPO consistently exhibits superior capabilities in two critical aspects: (1) precise task understanding and (2) concise and robust code generation. For instance, in the remove_duplicates task, SFT misconstrues the requirement by retaining the first occurrence of duplicates, leading to incorrect outputs. In contrast, InfiFPO accurately identifies the need to eliminate all repeated elements, employing a frequency-counting approach that aligns with the task's semantics. Similarly, in the vowels_count task, InfiFPO adopts a minimal yet effective implementation

| OpenAi HumanEval - **Task Type:** Code Generation | |
|---|---|
| **InfiFPO Behavior** | **Baseline Behavior** |

"prompt_1": "from typing import List\n\n\ndef remove_duplicates(numbers: List[int]) ->List[int]:\n \"\"\" From a list of integers, remove all elements that occur more than once.\n Keep order of elements left the same as in the input.\n >>>remove_duplicates([1, 2, 3, 2, 4])\n [1, 3, 4]\n \"\"\"\n"

| | |
|---|---|
| "completion":  "def remove_duplicates(numbers:  List[int]) ->List[int]:\n \"\"\" From a list of integers, remove all elements that occur more than once.\n Keep order of elements left the same as in the input.\n >>>remove_duplicates([1, 2, 3, 2, 4])\n [1, 3, 4]\n \"\"\"\n counts = {}\n for number in numbers:\n counts[number] = counts.get(number, 0) + 1\n \n result = []\n for number in numbers:\n if counts[number] == 1:\n result.append(number)\n \n return result\n","result":  "passed","passed": true | "completion":  "from typing import List\n\ndef remove_duplicates(numbers: List[int]) ->List[int]:\n \"\"\" From a list of integers, remove all elements that occur more than once.\n Keep order of elements left the same as in the input.\n >>>remove_duplicates([1, 2, 3, 2, 4])\n [1, 3, 4]\n \"\"\"\n seen = set()\n duplicates = set()\n result = []\n\n for number in numbers:\n if number in duplicates:\n continue\n if number in seen:\n duplicates.add(number)\n else:\n seen.add(number)\n result.append(number)\n\n return result\n","result":  "failed: ","passed":  false |

"prompt_2": "\nFIX = \"\"\"\nAdd more test cases.\n\"\"\"\n\ndef vowels_count(s):\n \"\"\"Write a function vowels_count which takes a string representing\n a word as input and returns the number of vowels in the string.\n Vowels in this case are 'a', 'e', 'i', 'o', 'u'. Here, 'y' is also a\n vowel, but only when it is at the end of the given word.\n\n Example:\n >>>vowels_count(\"abcde\")\n 2\n >>>vowels_count(\"ACEDY\")\n 3\n \"\"\"\n"

| | |
|---|---|
| "completion":  "def vowels_count(s):\n \"\"\"Write a function vowels_count which takes a string representing\n a word as input and returns the number of vowels in the string.\n Vowels in this case are 'a', 'e', 'i', 'o', 'u'.  Here, 'y' is also a\n vowel, but only when it is at the end of the given word.\n\n Example:\n >>>vowels_count(\"abcde\")\n 2\n >>>vowels_count(\"ACEDY\")\n 3\n \"\"\"\n vowels = \"aeiouAEIOU\"\n count = 0\n for char in s:\n if char in vowels:\n count += 1\n if s[-1].lower() == 'y':\n count += 1\n return count\n", "result":  "passed", "passed":  true | "completion":  "def vowels_count(s):\n \"\"\"Write a function vowels_count which takes a string representing\n a word as input and returns the number of vowels in the string.\n Vowels in this case are 'a', 'e', 'i', 'o', 'u'.  Here, 'y' is also a\n vowel, but only when it is at the end of the given word.\n\n Example:\n >>>vowels_count(\"abcde\")\n 2\n >>>vowels_count(\"ACEDY\")\n 3\n \"\"\"\n vowels = \"aeiouAEIOU\"\n count = 0\n for i, char in enumerate(s):\n if char in vowels:\n count += 1\n elif char == 'y' and i == len(s) - 1:\n count += 1\n return count\n\n# Additional test cases\nprint(vowels_count(\"hello\")) # 2\nprint(vowels_count(\"sky\")) # 1\nprint(vowels_count(\"rhythm\")) # 0\nprint(vowels_count(\"fly\")) # 1\nprint(vowels_count(\"boy\")) # 2\nprint(vowels_count(\"try\")) # 1\nprint(vowels_count(\"my\")) # 1\nprint(vowels_count(\"y\")) # 1\nprint(vowels_count(\"a\")) # 1\nprint(vowels_count(\"e\")) # 1\nprint(vowels_count(\"i\")) # 1\nprint(vowels_count(\"o\")) # 1\nprint(vowels_count(\"u\")) # 1\n", "result":  "failed:  Test 5","passed": false, |

Table 15: Case study on code generation.

that avoids boundary errors, whereas SFT introduces unnecessary loop-index conditions prone to edge-case failures.

These results suggest that InfiFPO is not merely replicating seen patterns but actively refining the reasoning chain to bridge the gap between instruction semantics and executable code. By reducing spurious complexity and enhancing robustness against implicit corner cases, InfiFPO demonstrates a stronger alignment with real-world coding practices.

## F.2 Mathematical Reasoning Perspective

| Case | Problem Prompt | SFT Output | FPO Output | Gold | Comment |
|------|----------------|------------|------------|------|---------|
| 0 | `Problem:` How many ways are there to divide a set of 8 elements into 5 non-empty ordered subsets? Please analyze and write your final answer after the phrase "The answer is": | SFT predicts 126000, by mistakenly recalling $S(8,5)$ and applying 5! permutations, leading to a significant overestimate. | InfiFPO correctly computes $S(8,5) = 105$, applies $5! = 120$, yielding $105 \times 120 = 12600$. | 11760 | InfiFPO demonstrates recursive computation of Stirling numbers, while SFT simply recalls a wrong value, lacking procedural reasoning. |
| 1 | `Problem:` What is the value of $\int_{-\infty}^{+\infty} \sin(3t) \cdot \sin(t/\pi)/t^2\, dt$? Please analyze and write your final answer after the phrase "The answer is": | SFT predicts 0, oversimplifying based on oscillation intuition, neglecting the structure of the integral. | InfiFPO expands product of sines, applies Fourier integral properties, identifies nonzero singular contribution, yielding the correct result. | 1.0 | InfiFPO's success stems from correct application of orthogonality and singularity analysis, while SFT lacks symbolic manipulation depth. |
| 2 | `Problem:` Given x = 0.157, what is the value of $x \times \frac{\prod_{n=1}^{\infty}(1 - \frac{x^2}{n^2\pi^2})}{\sin(x)}$? Please analyze and write your final answer after the phrase "The answer is": | Both SFT and InfiFPO predict 0.157, recognizing Euler's reflection formula, correctly simplifying numerator and denominator. | Both predict 0.157. FPO's derivation explicitly links product form to sine function identity. | 0.157 | Both models give correct answer, but InfiFPO provides a more explicit and pedagogical derivation path. |

Table 16: Comparison between SFT and InfiFPO on TheoremQA benchmark. InfiFPO shows stronger symbolic reasoning, procedural consistency, and structural understanding, leading to better robustness and interpretability.

We conduct a detailed comparison of InfiFPO and SFT on the TheoremQA benchmark to assess their mathematical reasoning capabilities. Our evaluation reveals that InfiFPO demonstrates superior performance in symbolic manipulation, stepwise reasoning, and the application of mathematical identities. While SFT often relies on pattern matching and heuristic recall, InfiFPO constructs solutions via complete, logically consistent derivation chains.

For instance, concerning the evaluation of the definite integral $\int_{-\infty}^{+\infty} \frac{\sin(3t)\sin(t/\pi)}{t^2}\, dt$, InfiFPO successfully decomposes the product of sine functions using trigonometric identities and correctly identifies the contribution of singularities and symmetry in determining the integral's value. In contrast, SFT prematurely concludes that the integral vanishes due to oscillation, neglecting deeper analysis of the integrand's behavior.

