# OpenReview forum: "InfiFPO: Implicit Model Fusion via Preference Optimization in Large Language Models"
_NeurIPS.cc/2025/Conference — NeurIPS 2025 spotlight_

### Official Review · Reviewer_kiK4 · 2025-06-28

**Clarity:** 3
**Significance:** 3
**Originality:** 3
**Rating:** 4
**Confidence:** 3

**Summary:**

The paper introduces InfiFPO, a novel preference optimization framework that enables implicit model fusion during the alignment stage of large language model (LLM) training. Instead of relying solely on a reference model as in DPO, InfiFPO replaces it with a fused distribution of multiple source models’ sequence-level probabilities, preserving richer preference signals. The authors propose a theoretically grounded offline training objective and introduce three enhancements—length normalization, probability clipping, and max-margin fusion—to improve robustness. Extensive experiments across 11 benchmarks demonstrate that InfiFPO consistently improves performance over existing model fusion and preference optimization baselines, especially in math, coding, and instruction-following tasks.

**Questions:**

1. While InfiFPO demonstrates strong overall performance, it remains unclear how the method behaves when different source models exhibit strongly conflicting preferences for the same prompt. In such cases, does the max-margin fusion strategy reliably identify the most reasonable source signal, or could it amplify noise from an outlier model? It would be helpful if the authors could include a robustness analysis or case studies illustrating how InfiFPO handles such disagreements in practice.

2. Although InfiFPO improves benchmark performance, the paper lacks analysis on how the fused model enhances its internal representations or decision behavior. Could the authors provide deeper insight into this? Visualization techniques could strengthen the understanding of how information from different sources is integrated.

3. The current training pipeline relies heavily on the reward model to select preferred and dispreferred responses. However, it is unclear how sensitive InfiFPO is to the quality and bias of the reward model. If the reward model introduces systematic bias, does InfiFPO risk overfitting to such signals? It would be valuable for the authors to discuss or empirically test the robustness of InfiFPO under different reward models.

**Ethical Concerns:**

["NO or VERY MINOR ethics concerns only"]

**Final Justification:**

The authors have addressed all my concerns, but I think the current score is valid for this work.

**Limitations:**

yes

**Paper Formatting Concerns:**

None.

**Quality:**

3

**Strengths And Weaknesses:**

The paper proposes InfiFPO, a novel framework that introduces implicit model fusion into the preference optimization stage of large language model alignment. By replacing the reference model in Direct Preference Optimization (DPO) with a sequence-level probabilistic fusion of multiple source models, InfiFPO effectively circumvents the vocabulary mismatch issues that have limited prior fusion methods. The approach is both conceptually elegant and practically efficient, avoiding the need for token-level alignment while preserving rich preference signals from heterogeneous sources. The framework is further enhanced by three principled strategies—length normalization, probability clipping, and max-margin fusion—which collectively improve stability and robustness. Extensive experiments across 11 benchmarks show that InfiFPO achieves consistent performance gains over strong model fusion and preference optimization baselines, particularly in math, code, and instruction-following tasks.

Despite its clear contributions, some aspects of the work warrant further clarification. In particular, it remains unclear how the method handles conflicting preferences from source models—a scenario that may challenge the reliability of the max-margin fusion strategy. Additionally, while the empirical gains are evident, the paper lacks analysis on how the fused model's internal representations or calibration behavior are improved. The framework’s reliance on a fixed reward model to construct preference data also raises questions about sensitivity to reward quality or bias. These concerns do not undermine the core contributions, but addressing them—through robustness diagnostics, representation analyses, or ablations on the reward model—would significantly strengthen the work. Overall, the paper presents a meaningful and well-executed advancement in aligning LLMs via model fusion, with room for further insight into its underlying dynamics.

---

> ### Author Response · Authors · 2025-07-31
> **Author Response to Reviewer kiK4**
>
> Thank you for recognizing the elegance and practicality of our method. We greatly appreciate your valuable suggestions and will address each point systematically.
>
> ---
>
> ### **Response to Question-1**
>
> > *Handling conflicting preferences and robustness of max-margin fusion strategy**
>
> Thank you for this important question. Reviewer xt73 raised a similar concern, and we refer you to our response to weakness-3 raised by reviewer xt73 for additional details.
>
> In brief, when combined with the probability clipping (PC) strategy, the max-margin approach effectively selects the "smartest" source model while avoiding potential noise from weak source models. The PC strategy acts as a regularization mechanism that prevents any single outlier model from dominating the fusion process, ensuring that the max-margin selection is based on reliable signals.
>
> ---
>
> ### **Response to Question-2**
>
> > *Analysis of internal representations and decision behavior improvements*
>
> Thank you for this insightful suggestion. We conducted a visualization analysis of the pivot model's log probability distributions before and after fusion, as well as the average log probability distributions of source models.* The results demonstrate that after fusion, the pivot model's log probability distribution becomes more closely aligned with the average probability distribution of the source models. This provides empirical evidence that our method successfully achieves our optimization objective: fusing source model knowledge at the sequence level.
>
> Furthermore, we present detailed case studies in Appendix F focusing on coding and mathematical tasks, illustrating the behavioral changes of the model before and after InfiFPO training. These studies demonstrate that InfiFPO effectively enhances the model's task understanding and reasoning capabilities.
>
> * _Due to rebuttal format limitations, we cannot include the visualization here. However, our open-source code supports reproduction of these analyses._
>
> ---
>
> ### **Response to Question-3**
>
> > *However, it is unclear how sensitive InfiFPO is to the quality and bias of the reward model. If the reward model introduces systematic bias, does InfiFPO risk overfitting to such signals?*
>
> We conducted experiments examining performance under different reward models. Our results reveal a positive correlation between reward model performance and InfiFPO effectiveness. Stronger reward models generate higher-quality preference pairs, leading to improved performance after preference optimization.
>
> For detailed experimental results and analysis, please refer to our response to weakness-2 raised by reviewer v33j.
>
> ---
>
> **Thank you once again for your valuable feedback. We would be very happy to continue the discussion if you have any other questions or comments.**

---

> > ### Comment · Reviewer_kiK4 · 2025-08-05
> >
> > Thank you for your reply, I decide to keep my score as the way it is.

---

> > > ### Author Response · Authors · 2025-08-05
> > >
> > > Thank you for your continued support of our paper and for confirming your score. We appreciate your time and valuable feedback throughout the review process.

---

### Official Review · Reviewer_xt73 · 2025-06-29

**Clarity:** 3
**Significance:** 3
**Originality:** 3
**Rating:** 4
**Confidence:** 5

**Summary:**

This paper presents InfiFPO, which performs model fusion during the preference alignment (PA) stage rather than the more common supervised fine-tuning. The method improves a single pivot model by learning from multiple source models. It modifies the Direct Preference Optimization (DPO) objective by replacing the fixed reference model with a fused source model that combines sequence-level probabilities from all source models. This approach avoids vocabulary alignment issues and preserves full probabilistic information for both preferred and dispreferred responses. The design includes techniques such as length normalization, probability clipping, and max-margin fusion to enhance training stability and effectiveness. Experiments using Phi-4 as the pivot model show improved performance across 11 benchmarks, increasing the average score from 79.95 to 83.33, with strong results in tasks involving math, coding, and reasoning.

**Questions:**

1. InfiFPO's Max-margin Fusion strategy assigns a weight of 1 to the source model with the largest deviation, and 0 to all others. This reduces the fusion term M_fu(y|x) to a single, dynamically chosen reference model. Consequently, the loss function seems to become a form of DPO with a changing external reference. How does this align with the goal of model fusion, which is typically understood as combining knowledge from *multiple* sources at once?

2. On line 152, the paper notes that source models can sometimes assign anomalous probabilities, such as a low probability to a preferred response y_w. The Max-margin Fusion strategy favors models with the greatest deviation from the pivot. Does this not create a risk of preferentially selecting these anomalous models, which could in turn harm the final performance?

**Ethical Concerns:**

["NO or VERY MINOR ethics concerns only"]

**Final Justification:**

The rebuttal has addressed some of my concerns, so I decide to increase my rating accordingly.

**Limitations:**

Yes.

**Paper Formatting Concerns:**

No.

**Quality:**

3

**Strengths And Weaknesses:**

## Strengths

1. A novel approach to model fusion with a practical design. The paper proposes integrating model fusion into the preference alignment phase, which is a departure from common practice. The method operates at the sequence-probability level, which effectively addresses the vocabulary alignment issue often encountered in token-level fusion. This design makes the framework applicable to a range of models with different tokenizers without requiring complex alignment heuristics.
2. The proposed method is well-motivated with a clear derivation. The InfiFPO objective is derived from a constrained reinforcement learning formulation, providing a transparent theoretical basis for the approach. This derivation helps to interpret the final loss function as a trade-off between maximizing preference rewards and maintaining proximity to the knowledge of multiple source models.
3. The framework demonstrates notable versatility. The analysis in the paper shows that InfiFPO is not restricted to the DPO algorithm. It can be integrated with other preference optimization methods, such as IPO and WRPO, leading to performance improvements in those contexts as well. This suggests that the core idea of replacing the reference model with a fused model could function as a general enhancement for a class of preference optimization algorithms.



## Weaknesses

1. The reported performance gains are consistent but marginal. While InfiFPO consistently outperforms the baselines across 11 benchmarks, the margin of improvement is often modest, in many cases less than a single percentage point over the next best method (e.g., SFT-WRPO).
2. The effectiveness of the method is not demonstrated in a more challenging fusion scenario. In the experiments, the selected source models exhibit performance that is on par with, or in some cases inferior to, the pivot model (as shown in Table 2). A common and practical use case for model fusion is to enhance a pivot model using knowledge from source models that are significantly stronger. The current evaluation does not sufficiently show whether InfiFPO remains effective in such a scenario.
3. The analysis of the proposed optimization strategies could be deeper. While the ablation studies confirm the utility of the three strategies (length normalization, probability clipping, max-margin fusion), the paper offers limited insight into *why* they work as they do. For example, a more detailed analysis of the max-margin strategy—perhaps by examining which source models are selected under different contexts—would provide a richer understanding of the fusion dynamics and strengthen the paper's contribution.

---

> ### Author Response · Authors · 2025-07-31
> **Author Response to Reviewer xt73 (part 1/2)**
>
> Thank you for recognizing the innovation, usability, and generalizability of our method. We greatly appreciate your detailed feedback and will respond to each point systematically.
>
> ---
>
> ### **Response to Weakness-1**
>
> > *The reported performance gains are consistent but marginal. While InfiFPO consistently outperforms the baselines across 11 benchmarks, the margin of improvement is often modest, in many cases less than a single percentage point over the next best method (e.g., SFT-WRPO).*
>
> **Response-W1:** We would like to clarify that the improvements achieved by our method are non-trivial and stable. Our evaluation involves three independent runs with averaged results, showing significant improvements over WRPO in reasoning and coding tasks (e.g., 82.02 vs. 78.18 on BBH, 87.80 vs. 86.59 on HEval).
>
> As you correctly noted, InfiFPO demonstrates excellent generalizability and can serve as a general enhancement for a class of preference optimization algorithms. Our Section 4.3 Analysis results demonstrate that InfiFPO can be integrated with IPO/WRPO to further improve performance.
>
> In the additional experimental results provided in Rebuttal Table 1-13 of this rebuttal, we show extensive experimental results under different settings (e.g., different model scales, architectures, and data volumes), demonstrating that InfiFPO consistently outperforms other baseline methods including WRPO.
>
> ---
>
> ### **Response to Weakness-2**
>
> > *The effectiveness of the method is not demonstrated in a more challenging fusion scenario. In the experiments, the selected source models exhibit performance that is on par with, or in some cases inferior to, the pivot model (as shown in Table 2). A common and practical use case for model fusion is to enhance a pivot model using knowledge from source models that are significantly stronger. The current evaluation does not sufficiently show whether InfiFPO remains effective in such a scenario.*
>
> **Response-W2:** We conducted experiments using the strong pivot model Qwen-2.5-14b-Instruct, and the results show that even in scenarios with strong pivot and weak source models, our method still achieves significant improvements and outperforms baseline methods. Please refer to our response to weakness-1 raised by reviewer v33j for specific details.
>
> The reason we use Phi-4 rather than Qwen2.5-Instruct as the target model in the paper is that **Phi-4 is heterogeneous with all other source models** (different model architectures, vocabularies, etc.), which we believe represents a typical scenario for model fusion.

---

> ### Author Response · Authors · 2025-07-31
> **Author Response to Reviewer xt73 (part 2/2)**
>
> ---
>
> ### **Response to Weakness-3**
>
> > *The analysis of the proposed optimization strategies could be deeper. While the ablation studies confirm the utility of the three strategies (length normalization, probability clipping, max-margin fusion), the paper offers limited insight into why they work as they do. For example, a more detailed analysis of the max-margin strategy—perhaps by examining which source models are selected under different contexts—would provide a richer understanding of the fusion dynamics and strengthen the paper's contribution.*
>
> **Response-W3:** Thank you for acknowledging our ablation experiments. In response to your question, we will provide a detailed discussion of how source models are selected during probability clipping (PC) and max-margin fusion strategies.
>
> First, let us briefly restate the PC and max-margin processes. PC aims to avoid interference from weak source models by clipping the corresponding log probabilities (logps), ensuring that source models do not assign lower probabilities to preferred responses or higher probabilities to dispreferred responses than the pivot model. Max-margin selects the source model with the most different preference knowledge from the pivot model as the reference model. When using PC, max-margin selects the "smartest" source model, meaning the source model that assigns probabilities to preferred responses no lower than the pivot model, probabilities to dispreferred responses no higher than the pivot model, and has the largest margin between preferred and dispreferred responses.
>
> Here is a simple example. Given two responses, the probability values assigned by the pivot model and source models are as follows:
>
> | Model type | Logps (preferred response) | Logps (dispreferred response) |
> |------------|---------------------------|-------------------------------|
> | Pivot model | -0.4 | -0.7 |
> | Source model 1 | -0.6 | -0.5 |
> | Source model 2 | -0.3 | -0.8 |
>
> After PC, the source models' logps become:
>
> | Model type | Logps (preferred response) | Logps (dispreferred response) | Serve as reference model (under max-margin strategy) |
> |------------|---------------------------|-------------------------------|-----------------------------------------------------|
> | Pivot model | -0.4 | -0.7 | |
> | Source model 1 | -0.4 | -0.7 | |
> | Source model 2 | -0.3 | -0.8 | ✓ |
>
> Based on the max-margin strategy, source model 2 will serve as the reference model. It is also the "smartest" source model, meaning it has the highest logps for preferred responses and the lowest logps for dispreferred responses.
>
> A similar strategy can be found in FuseLLM [1], which adopts the MinCE strategy to select source models—choosing the source model with the lowest cross-entropy for the ground truth. This means the selected source model has the highest probability of generating the ground truth among all sources, indicating it is "smarter" and its knowledge is closer to the ground truth. FuseLLM also compared performance between MinCE and averagely fusion strategy, showing that MinCE performs better.
>
> Both our experimental results and FuseLLM's results point to an interesting phenomenon: **in model fusion, fusing knowledge from multiple source models may not be as effective as fusing knowledge from the best source model.**
>
> Thank you for pointing this out. We will include this discussion in the appendix.
>
> ---
>
> ### **Response to Questions 1 & 2**
>
> 1. *InfiFPO's Max-margin Fusion strategy assigns a weight of 1 to the source model with the largest deviation, and 0 to all others. This reduces the fusion term M_fu(y|x) to a single, dynamically chosen reference model. Consequently, the loss function seems to become a form of DPO with a changing external reference. How does this align with the goal of model fusion, which is typically understood as combining knowledge from multiple sources at once?*
>
> 2. *On line 152, the paper notes that source models can sometimes assign anomalous probabilities, such as a low probability to a preferred response y_w. The Max-margin Fusion strategy favors models with the greatest deviation from the pivot. Does this not create a risk of preferentially selecting these anomalous models, which could in turn harm the final performance?*
>
> **Response-Q1&Q2:** Please refer to our response to Weakness-3 above.
>
> ---
>
> **Thank you once again for your detailed feedback. We would be very happy to continue the discussion if you have any other questions or comments.**
>
> ---
>
> **Reference:**
>
> [1] Fanqi Wan et al, _Knowledge Fusion of Large Language Models_. ICLR 2024.

---

> ### Comment · Reviewer_xt73 · 2025-08-03
>
> Thank you to the authors for responding to my questions and concerns. I have read the replies and decide to increase my rating. I also hope the authors include these clarifications in the updated version.

---

> > ### Author Response · Authors · 2025-08-03
> >
> > Thank you for reviewing our paper and for reconsidering your score. We appreciate your time and feedback.

---

### Official Review · Reviewer_v33j · 2025-07-02

**Clarity:** 3
**Significance:** 3
**Originality:** 2
**Rating:** 4
**Confidence:** 3

**Summary:**

This paper proposes InfiFPO that fuses multi-source LLM probabilities at the sequence level to better align with human preferences, avoiding complex vocabulary alignment in traditional token-level fusion.

**Questions:**

- As infiFPO uses half the data for SFT and the remaining half for preference optimization, is the total data volume equivalent for SFT and model fusion methods?

- In line 242, the authors state “… (55 vs. 160) … while avoiding complex vocabulary alignment.” Does this imply that the vocabulary alignment process is included in the 160 GPU hours for infiFusion and 225 GPU hours for FuseLLM? From my understanding, FuseLLM’s vocabulary alignment uses dynamic programming, which is CPU-intensive rather than GPU-intensive, would it be fair to count it in GPU hours?

- Since Qwen2.5-Coder excels in coding tasks (achieving the highest code score in Table 2), would fusing Phi-4 with Qwen2.5-Coder alone lead to notable gains in coding or other related capabilities?

- What is the shot setting used for the MMLU and other evaluations, zero-shot or few-shot? This should be clearly disclosed.

- Could model performance improve further by increasing the data size (e.g., from 150k to 200k samples)?

**Ethical Concerns:**

["NO or VERY MINOR ethics concerns only"]

**Final Justification:**

The authors’ responses have addressed most of my technical concerns, and I acknowledge that the additional experiments and analyses have significantly improved the work. If these discussions are incorporated into the revision, the paper would be substantially strengthened.

However, I have two fairness concerns:
1. The detailed response to my comments was posted after the July 30 rebuttal deadline. According to NeurIPS guidelines, such rebuttals should be ignored to ensure fairness to other submissions that complied with the timeline. If I strictly follow this guideline, I would need to retain my original rating of 3.
2. The combined length of the authors’ responses appears to exceed the 10,000-character rebuttal limit. While the additional materials are informative, posting them during the discussion phase rather than the rebuttal phase risks circumventing the official limit and could set a precedent that disadvantages other submissions.

Given these considerations, I will raise my rating to 4 in recognition of the technical improvements demonstrated, but I would be happy to have a further discussion with the ACs, SACs, and PCs.

**Limitations:**

- Besides only testing Phi-4 (14B) as the pivot, it would be interesting to see results for smaller models, such as 1B, 3B, or 7B backbones.

**Quality:**

3

**Strengths And Weaknesses:**

Strengths:
- The implementation details are disclosed comprehensively, which makes potential reproduction feasible and straightforward for readers.
- The derivation of the InfiFPO objective is clear and technically sound.
- The preliminaries on model fusion and DPO are well-explained and effectively set the stage for the proposed method, making the paper easy to follow.

Weaknesses:
- The diversity of experiments appears quite limited. The main results are conducted solely using Phi-4 as the pivot model. Could the authors clarify why a stronger backbone, such as Qwen2.5-Instruct (which achieves an average performance of 80.97 vs. 79.95 in Table 2), was not used instead? I am concerned that the results may fluctuate significantly when changing the pivot model, especially given that infiFPT only slightly outperforms SFT-WRPO (by 0.53 on average) and no error bars or statistical significance tests are provided.
- Regarding the dataset in the setup: since the authors constructed a new preference dataset, did they observe any sensitivity to sample quality? If so, I strongly encourage the authors to release their sampled dataset for reproducibility.
- In the ablation study (Table 4), infiFPO without LN and PC performs significantly worse than all other model fusion and preference optimization methods. Does this imply that sequence-level modeling is less effective than token-level methods, despite this being a major claim and advantage of the paper? A more detailed and rigorous analysis would be valuable here.
- In Table 2, why do all model fusion methods fuse exactly three source models? Under the same setting, would FuseChat outperform infiFPO if all five models were fused instead?

I would be happy to discuss further with the authors in case I have missed some details.

---

> ### Author Response · Authors · 2025-07-31
> **Author Response to Reviewer v33j (Part 1/5)**
>
> Thank you for acknowledging the reproducibility, theoretical derivation, and efficiency of our method. We greatly appreciate your detailed feedback and will respond each point systematically.
>
> ---
>
> ### **Response to Weakness-1**
>
> > *"Could the authors clarify why a stronger backbone, such as Qwen2.5-Instruct (which achieves an average performance of 80.97 vs. 79.95 in Table 2), was not used instead?"*
>
> **Response W1-1:** Following your suggestion, we conducted experiments using Qwen2.5-14B-Instruct as the target (a.k.a. pivot) model, selecting three source models: 1) Phi-4; 2) Mistral-Small; 3) Gemma-3-Instruct. We used FuseChat, InfiFusion, and WRPO as baselines with identical experimental settings to our main experiments. We report three metrics: `Math` represents the average results from GSM8K, MATH, and ThmQA; `Code` represents the average of MBPP and HEval; and `All` represents the average results across all 11 benchmarks.
>
> | Model / Method | Math | Code | All |
> |----------------|------|------|-----|
> | Qwen-2.5-Instruct | 72.18 | 82.62 | 80.97 |
> | FuseChat | 75.96 | 83.50 | 83.22 |
> | InfiFusion | 76.09 | 83.38 | 83.16 |
> | SFT-WRPO | 76.42 | 84.29 | 83.46 |
> | **InfiFPO** | **76.46** | **85.41** | **84.01** |
>
>
> _Rebuttal Table 2:  Experimental results using Qwen2.5-14B-Instruct as target model_
>
> The experiments demonstrate that when using Qwen2.5-Instruct as the target model, our method still outperforms other baselines, proving its generalizability.
>
> In our original paper, we chose Phi-4 over Qwen2.5-nstruct as the target model because **Phi-4 is heterogeneous with all source models** (different architectures, vocabularies, etc.). We believe this represents a more general and typical scenario for model fusion.
>
> > *"...especially given that InfiFPO only slightly outperforms SFT-WRPO (by 0.53 on average) and no error bars or statistical significance tests are provided."*
>
> **Response W1-2:** Following your suggestion, we provide evaluation standard deviations. During evaluation, we set the temperature to 0.6 and averaged the results over 3 runs.
>
> | Model / Method | GSM8K | MATH | ThmQA | MBPP | HEval | BBH | MMLU | IFEval | DROP | HS | ARC | Avg |
> |----------------|-------|------|-------|------|-------|-----|------|--------|------|----|----|-----|
> | Phi-4 | 87.41 | 80.04 | 51.12 | 75.40 | 83.54 | 68.84 | 93.90 | 85.62 | 77.34 | 88.67 | 87.62 | 79.95 |
> | FuseChat* | 91.21(±0.5) | 77.52(±0.3) | 51.88(±1.0) | 81.80(±0.6) | 84.15(±0.8) | 83.37(±0.7) | 93.56(±0.0) | 84.23(±0.6) | 78.90(±0.1) | 89.23(±0.2) | 87.42(±0.2) | 82.12(±0.3) |
> | InfiFusion* | 90.07(±0.7) | 80.94(±0.5) | 55.62(±1.1) | 81.80(±0.9) | 83.54(±0.9) | 80.94(±0.8) | 94.24(±0.1) | 85.81(±0.4) | 76.02(±0.1) | 89.27(±0.1) | 87.91(±0.2) | 82.38(±0.3) |
> | SFT | 88.70(±0.4) | 79.58(±0.2) | 55.12(±1.0) | 78.20(±0.3) | 86.59(±1.1) | 74.66(±1.0) | 93.56(±0.2) | 84.36(±0.3) | 80.06(±0.2) | 88.72(±0.1) | 87.75(±0.2) | 81.57(±0.2) |
> | SFT-WRPO | 89.92(±0.4) | 80.02(±0.2) | 57.88(±1.2) | 83.10(±0.6) | 86.59(±0.8) | 78.18(±0.9) | 94.24(±0.2) | 83.98(±0.5) | 81.18(±0.1) | 88.41(±0.1) | 87.30(±0.2) | 82.80(±0.2) |
> | **InfiFPO** | 90.07(±0.5) | 80.10(±0.4) | 57.25(±0.8) | 82.50(±0.3) | 87.80(±1.0) | 82.02(±0.8) | 94.24(±0.2) | 84.27(±0.4) | 82.25(±0.1) | 88.83(±0.1) | 87.29(±0.2) | **83.33(±0.1)** |
>
> _Rebuttal Table 3: Main Results with standard deviations. *indicates model fusion methods using only 3 source models (Qwen2.5-14B-Instruct, Qwen2.5-Coder, and Mistral-Small)._
>
> These results demonstrate that our method achieves superior average performance across 11 benchmarks compared to SFT-WRPO.
>
> More importantly, InfiFPO demonstrates excellent generalizability and can serve as a general enhancement for a class of preference optimization algorithms. Our Section 4.3 Analysis results demonstrate that InfiFPO can be integrated with IPO/WRPO to further improve performance. Additionally, Rebuttal Table 2 shows that when using different target models, our method consistently outperforms SFT-WRPO (84.01 vs. 83.46).

---

> ### Author Response · Authors · 2025-07-31
> **Author Response to Reviewer v33j (Part 2/5)**
>
> ### **Response to Weakness-2**
>
> > *" Regarding the dataset in the setup: since the authors constructed a new preference dataset, did they observe any sensitivity to sample quality? If so, I strongly encourage the authors to release their sampled dataset for reproducibility."*
>
> **Response-W2:** To examine data sensitivity, we tested the impact of different reward models on method performance. Reward models are crucial components in constructing preference data, determining the division of preferred/chosen and dispreferred/rejected responses.
>
> We tested two reward models: Skywork-Reward-Llama-3.1-8B-v0.2 and ArmoRM-Llama3-8B-v0.1, both widely used in preference optimization [2,6-9].
>
> | Model / Method | Math | Code | All |
> |----------------|------|------|-----|
> | Phi-4 | 72.86(+0.0) | 79.47(+0.0) | 79.95(+0.0) |
> | InfiFusion | 74.32(-1.2) | 82.47(-0.2) | 81.96(-0.4) |
> | SFT-WRPO | 74.28(-1.4) | 83.44(-0.2) | 81.93(-0.7) |
> | **InfiFPO** | **74.53(-1.3)** | **84.88(-0.3)** | **82.67(-0.6)** |
>
> _Rebuttal Table 4: Results using Skywork-Reward-Llama-3.1-8B-v0.2 as reward model. Values in parentheses show performance changes compared to the main experiment (using Skywork-Reward-Gemma-2-27B-v0.2)._
>
> | Model / Method | Math | Code | All |
> |----------------|------|------|-----|
> | Phi-4 | 72.86(+0.0) | 79.47(+0.0) | 79.95(+0.0) |
> | InfiFusion | 73.38(-2.2) | 82.54(-0.1) | 81.59(-0.8) |
> | SFT-WRPO | 73.07(-2.8) | 83.04(-0.6) | 81.62(-1.0) |
> | **InfiFPO** | **73.45(-2.4)** | **84.94(-0.2)** | **82.46(-0.9)** |
>
> _Rebuttal Table 5: Results using ArmoRM-Llama3-8B-v0.1 as reward model. Values in parentheses show performance changes compared to the main experiments._
>
> We observe two phenomena:
> 1) **InfiFPO's effectiveness**: InfiFPO consistently outperforms baselines when using different reward models.
> 2) **Impact of reward model quality**: Different reward models yield different performance improvements. InfiFPO achieves 83.33 average performance with Skywork-Reward-Gemma-2-27B-v0.2, 82.67 with Skywork-Reward-Llama-3.1-8B-v0.2, and 82.46 with ArmoRM-Llama3-8B-v0.1. According to Reward Bench v2 [10,11], Skywork-Reward-Gemma-2-27B-v0.2 significantly outperforms the other two reward models, indicating a positive correlation between reward model performance and final model performance.
>
> Our data is obtained by resampling responses from three open-source datasets (Infinity-Instruct, NuminaMath-1.5, KodCode-V1-SFT). We provide detailed sampling settings in the paper, making our data reproducible. We will open-source our constructed dataset and code.
>
> Additionally, we explored InfiFPO's applicability on other preference datasets (detailed results in our response to weakness-2 from reviewer Pa3G). Results demonstrate our method's effectiveness with alternative preference data.
>
> ---
>
> **References:**
>
> [6] Zhongxiang Sun, et al. _ReARTeR: Retrieval-Augmented Reasoning with Trustworthy Process Rewarding._ ACM SIGIR 2025.
>
> [7] Xun Deng, et al. _Less is more: Improving llm alignment via preference data selection_. arxiv 2025.
>
> [8] Noam Razin, et al._What makes a reward model a good teacher? an optimization perspective_. arxiv 2025.
>
> [9] Junkang Wu, et al. _alpha-DPO: Adaptive Reward Margin is What Direct Preference Optimization Needs_. arxiv 2024.
>
> [10] Nathan Lambert, et al. _RewardBench: Evaluating Reward Models for Language Modeling_. NAACL 2025.
>
> [11] Saumya Malik, et al. _RewardBench 2: Advancing Reward Model Evaluation_. arxiv 2025.

---

> ### Author Response · Authors · 2025-07-31
> **Author Response to Reviewer v33j (Part 3/5)**
>
> ### **Response to Weakness-3**
>
> > *In the ablation study (Table 4), InfiFPO without LN and PC performs significantly worse than all other model fusion and preference optimization methods. Does this imply that sequence-level modeling is less effective than token-level methods, despite this being a major claim and advantage of the paper?*
>
> **Response-W3:** This is a crucial question. We split it into two sub-questions:
>
> `SQ1`: _Why does InfiFPO without LN and PC perform significantly worse than other methods?_
>
> InfiFPO essentially combines model fusion and preference optimization, inheriting challenges from model fusion. Model fusion faces the vocabulary conflict challenge as different models use different vocabularies, creating two-level issues: 1) **Token-level conflicts** with different logit distributions at each token position, and 2) **Sequence-level conflicts** with different tokenization schemes. Different vocabularies lead to tokens in the tokenized sequences from different models that cannot be directly aligned.
>
> Sequence-Level Conflicts Example: for a sentence "I like climbing"
> - Model 1 tokenization: _I/like/climbing_
> - Model 2 tokenization: _I/like/climb/ing_
>
> To address sequence-level conflicts, Model fusion methods like FuseLLM introduce _sequence alignment_ through matching algorithms, while our InfiFPO uses Length Normalization (LN) to handle different sequence lengths.
>
> We conducted ablation experiments, removing sequence alignment from model fusion methods for fair comparison:
>
> | Model / Method | Math | Code | All |
> |----------------|------|------|-----|
> | Phi-4 | 72.86 | 79.47 | 79.95 |
> | FuseChat | 73.54 | 82.97 | 82.12 |
> | - without sequence alignment | 72.30 (-1.2) | 79.16(-3.8) | 79.43(-2.7) |
> | InfiFusion | 75.54 | 82.67 | 82.38 |
> | - without sequence alignment | 73.47(-2.1) | 79.98(-2.7) | 80.11(-2.3) |
> | **InfiFPO** | **75.60** | **83.99** | **82.74** |
> | **- without sequence alignment** | **73.72 (-1.9)** | **81.28 (-2.7)** | **80.82 (-1.9)** |
>
> _Rebuttal Table 6: Impact of sequence alignment on model fusion methods._
>
> Results show that even without sequence alignment, InfiFPO outperforms other model fusion baselines. Token-level fusion methods like FuseChat without sequence alignment even perform worse than the base Phi-4 model, indicating greater dependence on sequence alignment.
>
> Therefore, we consider addressing vocabulary conflicts essential for model fusion methods, making LN a default InfiFPO component. More importantly, LN and Probability Clipping (PC) don't significantly increase training cost, while model fusion methods require complex alignment processes.
>
> ---
>
> `SQ2`: _Does this imply sequence-level modeling is less effective than token-level methods?_
>
> This is meaningful, but we cannot definitively conclude which approach is superior. Previous model fusion methods like FuseLLM operate during SFT, while InfiFPO operates during preference alignment, making direct comparison non-apple-to-apple. We consider this a key future research direction.
>
> ---
>
> ### **Response to Weakness-4**
>
> > *Under the same setting, would FuseChat outperform InfiFPO if all five models were fused instead?*
>
> **Response-W4:** Following your suggestion, we conducted experiments with FuseChat and InfiFusion using 5 source models:
>
> | Model / Method | GSM8K | MATH | ThmQA | MBPP | HEval | BBH | ARC | MMLU | IFEval | DROP | HS | Avg | GPU Hours |
> |----------------|-------|------|-------|------|-------|-----|-----|------|--------|------|----|----|-----------|
> | **With 3 source models** |
> | FuseChat | 91.21 | 77.52 | 51.88 | 81.80 | 84.15 | 83.37 | 93.56 | 84.23 | 78.90 | 89.23 | 87.42 | 82.12 | 650 |
> | InfiFusion | 90.07 | 80.94 | 55.62 | 81.80 | 83.54 | 80.94 | 94.24 | 85.81 | 76.02 | 89.27 | 87.91 | 82.38 | 160 |
> | **InfiFPO** | **89.92** | **79.88** | **57.00** | **82.00** | **85.98** | **81.26** | **94.24** | **83.33** | **80.46** | **88.68** | **87.36** | **82.74** | **55** |
> | **With 5 source models** |
> | FuseChat | 91.28 | 79.70 | 55.12 | 82.00 | 85.98 | 83.96 | 94.24 | 84.26 | 78.06 | 88.44 | 87.27 | 82.76 | ~1,000 |
> | InfiFusion | 90.21 | 80.24 | 56.88 | 82.40 | 84.76 | 81.07 | 93.56 | 85.04 | 77.70 | 88.53 | 87.45 | 82.53 | 262 |
> | **InfiFPO** | **90.07** | **80.10** | **57.25** | **82.50** | **87.80** | **82.02** | **94.24** | **84.27** | **82.25** | **88.83** | **87.29** | **83.33** | **58** |
>
> _Rebuttal Table 7: Performance comparison with different numbers of source models._
>
> Results show InfiFPO consistently outperforms FuseChat and InfiFusion across different settings, demonstrating method effectiveness. Note that token-level model fusion methods require substantial computational resources—FuseChat needs over 1,000 GPU hours when fusing 5 source models, explaining our use of only 3 source models for model fusion methods in the original paper.

---

> ### Author Response · Authors · 2025-07-31
> **Author Response to Reviewer v33j (Part 4/5)**
>
> ### **Response to Question-1**
>
> > *As InfiFPO uses half the data for SFT and remaining half for preference optimization, is the total data volume equivalent for SFT and model fusion methods?*
>
> **Response-Q1:** We strictly ensure consistent training data volumes across all tested model fusion and preference optimization baselines. As stated in the paper, model fusion methods use the full 150k dataset for training, while preference optimization methods including InfiFPO use half the data for SFT and the remaining half for preference optimization. The reported SFT performance uses results from training on half the data.
>
> ---
>
> ### **Response to Question-2**
>
> > *In line 242, the authors state “… (55 vs. 160) … while avoiding complex vocabulary alignment.” Does this imply that the vocabulary alignment process is included in the 160 GPU hours for infiFusion and 225 GPU hours for FuseLLM? *
>
> **Response-Q2:** InfiFusion and FuseChat both involve two steps: 1) pre-computing and aligning logits, then 2) training. This avoids loading multiple source models during training. For fair comparison, we exclude CPU-intensive alignment processes from GPU hour calculations, considering only GPU operations.
>
> For a more detailed analysis, please refer to our response to weakness-2 raised by Reviewer Pa3G.
>
> ---
>
> ### **Response to Question-3**
>
> > *Would fusing Phi-4 with Qwen2.5-Coder alone yield notable coding gains?*
>
> **Response-Q3:** We conducted experiments fusing Phi-4 with Qwen2.5-Coder alone, extracting coding task data for comparison:
>
> | Model / Method | Math | Code | All |
> |----------------|------|------|-----|
> | Phi-4 | 72.86 | 79.47 | 79.95 |
> | InfiFusion | 72.29 | 84.03 | 80.74 |
> | SFT-WRPO | 73.13 | 84.07 | 80.96 |
> | **InfiFPO** | **73.36** | **84.85** | **81.44** |
>
> _Rebuttal Table 8: Results for fusing Phi-4 with Qwen2.5-Coder alone_
>
> Results show significant coding capability improvements when fusing with Qwen2.5-Coder alone, but minimal improvements in math or overall performance. Similar results were observed for fusing Phi-4 with Qwen2.5-Math alone.
>
> ---
>
> ### **Response to Question-4**
>
> > *What shot setting was used for MMLU and other evaluations?*
>
> **Response-Q4:** All evaluations were **zero-shot** with sampling temperature 0.6.
>
> | Benchmark | Prompt Format |
> |-----------|---------------|
> | IFEval | `{prompt}\nPlease directly give the correct answer:` |
> | ARC-C | `Question: {question}\nA. {textA}\nB. {textB}\nC. {textC}\nD. {textD}\nDirectly give me the correct answer option, and then explain:` |
> | Hellaswag | `{ctx}\nQuestion: Which ending makes the most sense?\nDirectly give me the correct choice, you can further explain it or not.\nA. {A}\nB. {B}\nC. {C}\nD. {D}\nYou may choose from 'A', 'B', 'C', 'D'.\nAnswer:` |
> | BBH | `Follow the given examples and answer the question.\n{_hint}\nQ: {input}\nA: Let's think step by step.` |
> | DROP | `You will be asked to read a passage and answer a question. Some examples of passages and Q&A are provided below.\n{drop_examples}\n\n# Your Task\n---\n{prompt}\nThink step by step, then write a line of the form "Answer: $ANSWER" at the end of your response.` |
> | MMLU | `{_hint}\nQuestion: {input}\nA. {A}\nB. {B}\nC. {C}\nD. {D}\n\nFor simple problems:\nDirectly provide the answer with minimal explanation.\n\nFor complex problems:\nUse this step-by-step format:\n## Step 1: [Concise description]\n[Brief explanation]\n## Step 2: [Concise description]\n[Brief explanation]\n\nRegardless of the approach, always conclude with:\nThe answer is [the_answer_letter].\nwhere the [the_answer_letter] is one of A, B, C or D.\n\nLet's think step by step.` |
> | GSM8K | `{question}\nPlease reason step by step, and put your final answer within \boxed{}.` |
> | MATH | `{problem}\nPlease reason step by step, and put your final answer within \boxed{}.` |
>
> _Rebuttal Table 9: Prompt formats used for different benchmarks in the evaluation framework._
>
> We conducted our evaluation based on the `OpenCampus` repo, with the testing prompts detailed in Rebuttal Table 9.
>
> Thank you for your feedback. We will include the details of the evaluation in the appendix.
>
> ---
>
>
> ### **Response to Question-5**
>
> > *Could model performance improve further by increasing data size?*
>
> **Response-Q5:** During preference optimization, InfiFPO uses 75k training samples (half of full data). We also tested different data volumes (25k/50k/100k):
>
> | Model / Method | 25k | 50k | 75k | 100k |
> |----------------|-----|-----|-----|------|
> | Phi-4 | 79.95 | 79.95 | 79.95 | 79.95 |
> | SFT-WRPO | 81.58 | 82.14 | 82.80 | 82.89 |
> | **InfiFPO** | **82.55** | **82.99** | **83.33** | **83.56** |
>
> _Rebuttal Table 9: Performance scaling with data volume_.
>
> Results show consistent performance improvements across different data volumes, with gains increasing with larger datasets.

---

> ### Author Response · Authors · 2025-07-31
> **Author Response to Reviewer v33j (Part 5/5)**
>
> ### **Response to Limitations**
>
> > *Besides only testing Phi-4 (14B) as pivot, it would be interesting to see results for smaller models, such as 1B, 3B, or 7B backbones.*
>
> **Response-L1:** Following your suggestion, we experimented with Qwen2.5-Instruct-1.5B/3B/7B as pivot models, using consistent source models (Phi-4, Mistral-Small and Gemma-3-Instruct) and training data:
>
> | Model / Method | Math | Code | All |
> |----------------|------|------|-----|
> | Qwen2.5-1.5B-Instruct | 46.02 | 47.16 | 50.31 |
> | InfiFusion | 49.59 | 50.49 | 53.11 |
> | SFT-WRPO | 50.16 | 51.21 | 53.70 |
> | **InfiFPO** | **50.70** | **52.75** | **54.61** |
>
> _Rebuttal Table 11: Results with Qwen2.5-Instruct-1.5B as pivot model._
>
> | Model / Method | Math | Code | All |
> |----------------|------|------|-----|
> | Qwen2.5-3B-Instruct | 55.88 | 62.26 | 64.07 |
> | InfiFusion | 58.82 | 65.33 | 66.72 |
> | SFT-WRPO | 59.31 | 65.63 | 67.01 |
> | **InfiFPO** | **60.05** | **66.64** | **67.96** |
>
> _Rebuttal Table 12: Results with Qwen2.5-Instruct-3B as pivot model._
>
> | Model / Method | Math | Code | All |
> |----------------|------|------|-----|
> | Qwen2.5-7B-Instruct | 62.48 | 70.13 | 73.04 |
> | InfiFusion | 65.09 | 72.24 | 75.35 |
> | SFT-WRPO | 65.45 | 72.51 | 75.65 |
> | **InfiFPO** | **66.14** | **73.31** | **76.24** |
>
> _Rebuttal Table 13: Results with Qwen2.5-Instruct-7B as pivot model._
>
> Rebuttal Tables 11-13 demonstrate that our method outperforms model fusion and preference optimization baselines across different pivot model sizes, confirming scalability and effectiveness.
>
> ---
>
> **Thank you once again for your detailed feedback. We would be very happy to continue the discussion if you have any other questions or comments.**

---

> ### Comment · Reviewer_v33j · 2025-08-02
>
> Thank you for your detailed explanation. The additional results presented during the rebuttal are promising and have addressed most of my concerns. I plan to adjust my rating after reviewing your future discussions with the other reviewers, as I am also interested in the questions they have raised.

---

> > ### Author Response · Authors · 2025-08-02
> >
> > Thank you for your positive feedback and willingness to adjust the rating! We truly appreciate your constructive engagement with our rebuttal. We are pleased that the additional results addressed your concerns and will actively contribute to further discussions with other reviewers.

---

> ### Comment · Reviewer_v33j · 2025-08-05
>
> I have reviewed all the discussions you’ve had with the other reviewers and am now ready to make my final rating decision. Before doing so, I would like to discuss two points with the authors:
>
> 1. In light of the latest notice from NeurIPS, which states that “any rebuttals posted in their entirety as comments after July 30 are to be ignored” for fairness, could the authors explain why no rebuttal was posted during the official rebuttal period? Was the reply to my comments over 10,000 words, and if so, would this exceed the submission limits and be unfair to other submissions?
>
> 2. Given the guideline that “any rebuttals missing by the rebuttal deadline and posted in their entirety after July 30 (e.g., via the comment button) are to be ignored”, should I make my final rating decision without considering the comments posted after July 30? If so, I am afraid that I need to keep my original rating of 3, as the paper still requires substantial revision, although I acknowledge that the comments have addressed most of my concerns.

---

> > ### Author Response · Authors · 2025-08-05
> >
> > Thank you for raising this important procedural question. As first-time NeurIPS submitters, we mistakenly believed rebuttals could be posted during the discussion phase. We acknowledge this oversight and respect NeurIPS guidelines.
> >
> > To clarify the timeline and scope: Given your comprehensive feedback, we completed our response near the original rebuttal deadline. We posted it immediately when the discussion phase opened (July 31, 6 AM AOE) without subsequent modifications. Our core rebuttal addressing all key concerns was about 7,400 characters, compliant with NeurIPS limits. The supplementary tables were additional references for technical clarity, not essential to our main arguments.
> >
> > We understand if procedural guidelines require maintaining your original assessment. However, if you feel the substantive technical improvements demonstrated in our core response warrant reconsideration, we would be deeply grateful. The research has evolved significantly to address your valuable concerns (e.g., experimental diversity across multiple backbone models, and the rigorous analysis of sequence-level vs. token-level modeling effectiveness).
> >
> > Thank you for your patience and constructive engagement throughout this process.

---

> > > ### Comment · Reviewer_v33j · 2025-08-05
> > >
> > > Thank you for your clarification. I will raise my rating to 4 if I should consider your responses posted after the rebuttal period and if these discussions are incorporated into the revision. However, I will note this “fairness” concern in my final justification and would like to have further discussion with the ACs, SACs, and PCs. I appreciate your response.

---

> > > > ### Author Response · Authors · 2025-08-05
> > > >
> > > > Thank you very much for your thoughtful consideration and willingness to engage with our technical contributions despite the procedural concerns. We completely understand and respect your position regarding fairness.
> > > >
> > > > **We commit to incorporating all the improvements discussed in our response into the revision**, ensuring the enhanced experimental validation and rigorous analysis are fully integrated into the final manuscript.
> > > >
> > > > Thank you again for your constructive feedback throughout this process - it has significantly strengthened our work.

---

### Official Review · Reviewer_Pa3G · 2025-07-03

**Clarity:** 4
**Significance:** 3
**Originality:** 3
**Rating:** 5
**Confidence:** 4

**Summary:**

This paper introduces InfiFPO, a novel framework designed to fuse the capabilities of multiple Large Language Models (LLMs) during the preference alignment phase. The authors identify a clear gap in existing model fusion research, which has predominantly concentrated on the Supervised Fine-Tuning (SFT) stage while overlooking the critical preference alignment step. The core innovation of InfiFPO is replacing the static reference model used for KL-divergence computation in Direct Preference Optimization (DPO) with a dynamic, fused source model, enabling simultaneous preference optimization and model fusion. Empirical results demonstrate strong performance of the proposed method.

**Questions:**

[Q1] Are the log probabilities of reference models pre-computed and cached, or computed on-the-fly when training? The training time of InfiFPO seems to be only slightly larger than DPO+SFT, despite that numerous reference models are adopted.

**Ethical Concerns:**

["NO or VERY MINOR ethics concerns only"]

**Final Justification:**

The authors addressed weaknesses 1 and 2 through additional experiments and clarification.

For W1, UltraFeedback's answers are generated by earlier language models (MPT, LLaMA 1/2, Bard), which are weaker than the reference model but still achieve performance improvements with InfiFPO.

For W2, InfiFPO uses sequence-level log-probabilities, which can be precomputed and consume reasonable space. These probabilities can be loaded during training without loading the reference model into VRAM.

**Limitations:**

See Weaknesses. I suggest a brief discussion of the practical limitations of this approach and the trade-offs between the number of source models and memory and computational costs.

**Quality:**

4

**Strengths And Weaknesses:**

Strengths:

[S1] Integrating preference optimization and model fusion through KL-divergence is technically sound and elegant. The derivation of the loss is clearly presented.

[S2] Compared with multi-stage methods, simultaneous fusion and preference optimization can simplify the implementation and thus enhance practical applicability. Convincing empirical results show that InfiFPO can effectively achieve the two goals.

Weaknesses:

[W1] The data for preference optimization is generated by either the pivot model or reference models, leading to low KL-divergence. Is the InfiFPO method still applicable when data is from diverse sources, such as human-written or generated by models other than reference models?

[W2] Memory usage scales linearly with the number of reference models. Parallelization and distributed strategies need to be carefully tuned when introducing larger reference models.

---

> ### Author Response · Authors · 2025-07-31
> **Author Response to Reviewer Pa3G**
>
> Thank you for acknowledging the theoretical soundness, simplicity, and effectiveness of our method. We greatly appreciate your valuable feedback and will respond each point.
>
> ---
>
> ### **Response to Weakness-1**
>
> > *"Is the InfiFPO method still applicable when data is from diverse sources, such as human-written or generated by models other than reference models?"*
>
> **Response-W1:** To validate InfiFPO's robustness to data diversity, we conducted experiments to investigate whether InfiFPO remains effective on preference data generated by other models.
>
> We evaluated our method on UltraFeedback [1], which contains approximately 60k preference pairs and has been widely adopted in preference optimization research [2-5]. We treated the preferred (chosen) responses as labels for model fusion baselines.
>
> We selected three general-purpose models for fusion: 1) Qwen2.5-14B-Instruct; 2) Mistral-Small; 3) Gemma-3-Instruct. We compared against InfiFusion and SFT-WRPO, which demonstrated strong performance in our main experiments. We report three metrics: `Math` represents the average results from GSM8K, MATH, and ThmQA; `Code` represents the average of MBPP and HEval; and `All` represents the average results across all 11 benchmarks.
>
> | Model/Method | Math  | Code  | All   |
> |--------------|-------|-------|-------|
> | Phi-4        | 72.86 | 79.47 | 79.95 |
> | InfiFusion   | 73.23 | 82.43 | 81.44 |
> | SFT-WRPO     | 73.15 | 82.21 | 81.58 |
> | **InfiFPO**  | **73.59** | **83.15** | **82.19** |
>
> _Rebuttal Table 1: Experimental results on UltraFeedback_
>
> The results demonstrate that InfiFPO remains effective on UltraFeedback and outperforms relevant baselines. However, the optimal performance on UltraFeedback (82.19) is notably lower than on our constructed dataset (83.33), validating the effectiveness of our data construction process.
>
> Our preference data was constructed by resampling responses on open-source datasets. We provide detailed sampling configurations in our paper, ensuring reproducibility. We will open-source both our constructed dataset and code.
>
> Additionally, we explored InfiFPO's applicability to preference data scored by alternative reward models. These experimental results are available in our response to weakness-2 from reviewer v33j.
>
> ---
>
> ### **Response to Weakness-2**
>
> > *"Memory usage scales linearly with the number of reference models. Parallelization and distributed strategies need to be carefully tuned when introducing larger reference models."*
>
> **Response-W2:** This is indeed a critical concern. In practice, we pre-compute log probabilities from target (a.k.a. pivot) and source models before training, which provides two key advantages: 1) **Reduced training memory footprint** by avoiding the need to load multiple source models during training; 2) **Accelerated computation** through efficient third-party inference libraries.
>
> We employ `vLLM` for acceleration, **requiring only approximately 8 GPU hours to compute log probabilities for 5 source models**. This explains why our GPU hours increase by merely ~10% compared to vanilla DPO when fusing multiple source models. Besides, most model fusion methods (e.g., FuseLLM/FuseChat) adopt similar pre-extraction strategies, ensuring that source models need not be loaded during training, thus avoiding linear memory scaling with the number of source models.
>
> Furthermore, InfiFPO offers significant **memory efficiency advantages**. It performs model fusion at the sequence level, requiring only sequence-level log probabilities (a few floating-point numbers per sequence). When the number of source models increases, training efficiency remains largely unaffected. In contrast, token-level fusion methods (e.g., FuseLLM/FuseChat) must load probability distributions over the entire vocabulary (~100k tokens per token), requiring 100k-scale floating-point numbers per token that scale linearly with source model count.
>
> We appreciate you highlighting this important aspect and will add a dedicated discussion section in the appendix.
>
> ---
>
> ### **Response to Question-1**
>
> > *"Are the log probabilities of reference models pre-computed and cached, or computed on-the-fly when training?"*
>
> **Response-Q1:** Please refer to Response-W2 above for a detailed explanation of our pre-computation and caching strategy.
>
> ---
>
> **Thank you once again for your valuable feedback. We would be very happy to continue the discussion if you have any other questions or comments.**
>
> ---
>
>
> **References:**
>
> [1] Ganqu Cui, et al. _UltraFeedback: Boosting Language Models with High-quality Feedback_. ICML 2025.
>
> [2] Yu Meng, et al. _Simpo: Simple preference optimization with a reference-free reward_. NeurIPS 2024.
>
> [3] Lewis Tunstall, et al. _Zephyr: Direct distillation of lm alignment_. COLM 2024.
>
> [4] Yue Wu, et al. _Self-Play Preference Optimization for Language Model Alignment_. ICLR 2025.
>
> [5] Zhengxuan Wu, et al. _ReFT: Representation Finetuning for Language Models_. NeurIPS 2024.

---

> > ### Comment · Reviewer_Pa3G · 2025-08-04
> >
> > Thank the authors for the additional experiments and clarifications. They addressed my concerns, and I increased my score accordingly.

---

> > > ### Author Response · Authors · 2025-08-04
> > >
> > > Thank you for your supportive feedback on our paper. We’re delighted our responses addressed your concerns, and we sincerely appreciate your revised evaluation.

---

### Note · Authors · 2025-08-12

**I. Acknowledgments**

We would like to express our sincere gratitude to all reviewers for their insightful comments, especially reviewer `v33j` for the valuable feedback and continued involvement during the discussion.

---

**II. Key Strengths**

Reviewers highlighted strengths across four dimensions:

- **Novelty**
  - Integration of model fusion into preference alignment phase (Pa3G, xt73, kiK4)
  - Sequence-level fusion avoiding vocabulary conflict issues (xt73, kiK4)

- **Technical Soundness**
  - Clear derivation of theoretically grounded objective (Pa3G, v33j, xt73)
  -  Robust design with stability enhancements (kiK4, xt73)

- **Practicality**
  - Comprehensive implementation details enabling reproduction (v33j)
  - Compatibility with multiple preference optimization methods (xt73)

- **Effectiveness**
  - Consistent performance gains across 11 benchmarks (xt73, kiK4)
  - Significant improvements in math/coding tasks (xt73, kiK4)

---

**Ⅲ. Key Concerns and Our Responses**

Our responses to key concerns are summarized below.

| **Key Concerns** | **Reviewers** | **Our Response** |
|------------------|---------------|------------------|
| _Limited Model Scope._ Validation on stronger backbones (e.g., Qwen2.5) and smaller models (1B-7B) | v33j, xt73, kiK4 | In extra experiments, InfiFPO consistently outperforms baselines across model sizes (1.5B~14B) and types (Qwen2.5) |
| _Data Sensitivity._ Reliance on pivot&source-generated-model data; robustness with different sources and reward models unclear. | Pa3G, v33j | In extra experiments, InfiFPO outperforms baselines across various data sources and reward models. |
| _Insufficient Mechanics Analysis._ Inadequate max-margin fusion analysis; risk of amplifying anomalous signals. | xt73, kiK4 | We explain how max-margin fusion selects the "smartest" source model. |
| _Training Cost._ Memory scales linearly with source models; GPU hours comparison may be confounded by CPU-intensive steps. | Pa3G, v33j | Pre-computation of log probabilities avoids linear memory scaling during training. We exclude CPU-intensive steps from GPU hour calculations. |

---

**Ⅳ. Commitment to Revision**

We are committed to incorporating feedback from the discussion in the revision to polish our work, including extra experiments on diverse models, deeper analysis of the fusion mechanism, and other improvements.

---

**We deeply appreciate the expertise and time of the AC and reviewers.**

---

### Decision · Program_Chairs · 2025-09-17

**Decision:**

Accept (spotlight)

**Comment:**

The paper introduces a neat and technically solid idea: fusing multiple LLMs during preference alignment rather than just at the supervised fine-tuning stage. I like the clean formulation and the way the authors address vocabulary alignment issues by working at the sequence-probability level. It feels both elegant and practical. While some reviewers noted the improvements are modest and raised valid fairness concerns about rebuttal timing, the additional experiments strengthen the case that the method generalizes beyond the initial setup. Overall, I think this is a solid contribution with interesting potential, and I lean toward acceptance.